# An intein-split transactivator for intersectional neural imaging and optogenetic manipulation

Hao-Shan Chen [1,2,7], Xiao-Long Zhang[2,3,4,7], Rong-Rong Yang[1], Guang-Ling Wang[1], Xin-Yue Zhu[1,5], Yuan-Fang Xu[1], Dan-Yang Wang[1], Na Zhang[1], Shou Qiu[1,2], Li-Jie Zhan[1], Zhi-Ming Shen[1], Xiao-Hong Xu [1], Gang Long[3,4 ✉] & Chun Xu [1,2,6 ✉]

The cell-type-specific recording and manipulation is instrumental to disentangle causal neural mechanisms in physiology and behavior and increasingly requires intersectional control; however, current approaches are largely limited by the number of intersectional features, incompatibility of common effectors and insufficient gene expression. Here, we utilized the protein-splicing technique mediated by intervening sequences (intein) and devised an intein-based intersectional synthesis of transactivator (IBIST) to selectively control gene expression of common effectors in multiple-feature defined cell types in mice. We validated the specificity and sufficiency of IBIST to control fluorophores, optogenetic opsins and $Ca^{2+}$ indicators in various intersectional conditions. The IBIST-based $Ca^{2+}$ imaging showed that the IBIST can intersect five features and that hippocampal neurons tune differently to distinct emotional stimuli depending on the pattern of projection targets. Collectively, the IBIST multiplexes the capability to intersect cell-type features and controls common effectors to effectively regulate gene expression, monitor and manipulate neural activities.

[1] Institute of Neuroscience, State Key Laboratory of Neuroscience, Center for Excellence in Brain Science and Intelligence Technology, Chinese Academy of Sciences, Shanghai 200031, China. [2] University of the Chinese Academy of Sciences, Beijing 100049, China. [3] Institut Pasteur of Shanghai, Chinese Academy of Sciences, Shanghai 200031, China. [4] Beijing Advanced Innovation Center for Structural Biology, Tsinghua University, Beijing 100084, China. [5] School of Life Science and Technology, ShanghaiTech University, Shanghai 201210, China. [6] Shanghai Center for Brain Science and Brain-Inspired Intelligence Technology, Shanghai 201210, China. [7] These authors contributed equally: Hao-Shan Chen, Xiao-Long Zhang. ✉email: glong@ips.ac.cn; chun.xu@ion.ac.cn

The cell-type diversity is a hallmark of the brain circuitry that enables us to process sensory stimuli in the environment, generate appropriate behaviors and emotions and gain the knowledge from learned experiences. The properties of neural cell types play pivotal roles in governing the neural functions of circuit networks across brain regions[1–3], thus it becomes increasingly important for functional studies to specifically label a homogenous cell type depending on multiple features such as spatial locations, molecular markers and neuronal connections[3–5]. Genetic mouse lines offer great tools to intersectionally label specific neuronal population[6,7], but are often hindered by the requirement of high-level gene expression for optogenetic opsins and $Ca^{2+}$ indicators[6,8]. The breeding and crossing between multiple mouse lines can be laborious and time consuming, and region-specific control is also challenging. Combining the adeno-associated virus (AAV) vectors with transgenic mice brings powerful and convenient strategies to label a defined population in an intersectional manner[5] and drives sufficient expression of opsins and $Ca^{2+}$ indicators for optogenetic manipulation and neural recording, respectively[9]. Pioneering works have constructed AAV effectors for fluorophore and Channelrhodopsin-2 (ChR2) that selectively respond to specific combinations of controllers such as Cre and Flpo[4,10,11]. While more intersectional effectors are made available for optogenetic opsins and $Ca^{2+}$ indicators to match specific combinations of up to three controllers[12], the number of intersectional controllers/features is still limited. These effectors for opsins and $Ca^{2+}$ indicators, which demand high level of expressions, may exhibit varying expression levels depending on experimental conditions and hence require laborious functional validations. Thus, it is desirable to fulfill both the dependency on more controllers/features and the compatibility with common effectors that is widely applicable across brain regions and animal species.

To this end, we took a strategy of reconstituting a single controller to achieve the intersectional control[7,13]. This strategy supports multi-feature dependent reconstitution of the single controller and in turn enables the compatibility with common effectors, which could be well verified to have reliable and sufficient gene expressions for recording or manipulation. Because most existing cell-type-specific driver lines are controlled by Cre[14,15], we sought to reconstitute non-Cre controllers which can be complementary for Cre, and then exploited the application with AAV and rabies vectors which are broadly used in neural circuit studies. The tetracycline-controlled transactivator (tTA) is such a widely used controller and has additional features with temporal and reversible regulation. The tTA is formed by fusing tetracycline repressor protein with an activation domain of virion protein 16 (VP16) from herpes simplex virus[16,17]. We reasoned that splitting tTA into these two parts would achieve separable expressions in distinct constructs, which most likely result in the reconstitution of fully functional tTA. We leveraged the intervening sequences (intein) mediated protein splicing technique to ligate the split parts of tTA and achieve the efficient reconstitution of tTA. The intein has been shown to facilitate successful reconstitution of two parts together at the protein level in *Caenorhabditis elegans*[18], neural stem cell[19], cell line[20] and mouse in vivo[7]. Compared to other reconstitution approaches such as leucine zipper-mediated dimerization and adapter pair-based covalent bonding, the intein approach introduces minimal heterologous sequence and exhibits better kinetics and higher activity recovery[18,21–23].

In summary, we have developed a system with intein-based intersectional synthesis of transactivator (IBIST) to achieve the intersectional expression of a single controller. It in turn fulfills the compatibility with common effectors and multiplexes the intersectional features based on other existing controllers. This system supports: (1) flexible design of modular expression of split

tTA in various conditions, (2) compatibility with common effectors dependent on tetracycline-responsive promoter element (TRE), and (3) versatile use with the viral vectors and rich resources of increasing mouse driver lines[5,14,15,24]. To demonstrate the capability of IBIST to intersectionally control opsins and $Ca^{2+}$ indicators, we have created various AAV and rabies vectors to express tTA fragments controlled by site-specific recombinase, promoters or circuit tracing. We have validated the specificity and efficiency of IBIST tools both in vitro and in vivo, and then performed electrophysiological and photometric recording to demonstrate the successful application of IBIST tools for optogenetics and $Ca^{2+}$ imaging in various neuronal populations with multiple features.

## Results

**The design of IBIST and the validation in vitro and in vivo.** We split the tTA into N-terminal (Tet repressor, TetR) and C-terminal (three tandem minimal VP16 activation domains) parts from its structure junction[17]. The N-terminal part was further tagged with a blue fluorescence protein (BFP, Fig. 1a). To facilitate the tTA reconstitution from two split parts, we attached the N- and C- fragment of intein sequences of Gp41-1[18,21] to the N- and C-terminal parts of tTA because these intein sequences were reported to be more efficient and faster for protein splicing than other intein sequences[25,26]. This pair of ready-to-reconstitute subunits are referred as tTAN and tTAC respectively and are designed to form functional tTA after intein-mediate splicing (Fig. 1a–c).

To validate whether the tTA could be successfully reconstituted in mammalian cells, we expressed tTAN and tTAC as well as a TRE-controlled GFP reporter[27] in cultured HEK293T cells (see methods; TRE-dependent expression of histone-GFP, TVA and rabies glycoprotein, termed TRE-HTG). The fluorescence and protein level of TRE-dependent reporter became prominent after transfection, which was in stark contrast to that in control groups (Fig. 1d and f and Supplementary Fig. 1a). These results confirmed that split tTA are specifically rescued in the targeted cells with the help of intein-mediated splicing. All GFP-positive cells also expressed the fluorescence tag of tTAN (Fig. 1e and g), suggesting that the tTA reconstitution is highly specific.

To test whether similar strategy is applicable for reverse tTA (rtTA), we engineered the constructs for split rtTA with same intein sequences (Supplementary Fig. 1b). We first transfected the HEK293T cells with plasmids for C-terminal part of rtTA (rtTAC) and reporter TRE-mCherry. To test whether the rtTA fragments can be expressed by viral vectors for neural tracing, we then inoculated the cells with a rabies variant expressing the N-terminal part of rtTA and the GFP tag (termed RV-rtTAN), which could be a retrograde transsynaptic tracer[28]. In the presence of doxycycline, the fluorescence and protein level of TRE-mCherry became prominent after transfection, but non-detectable in the control group (Supplementary Fig. 1a, c and e). Nearly all mCherry positive cells were co-labeled by the RV-rtTAN (Supplementary Fig. 1d and f), suggesting that the rtTA reconstitution is highly specific. Taken together, the fragments of tTA and rtTA are both able to efficiently and specifically reconstitute tTA and rtTA in the mammalian cells with the help of intein sequences, respectively.

To validate whether the split tTA could form functional reconstitution in vivo, we produced the AAV vectors and injected into the hippocampus with the AAVs of TRE-HTG and CaMKIIα dependent tTAC and BFP tagged tTAN. We observed strong fluorescence expression of reporter AAV-TRE-HTG in the hippocampus, whereas little fluorescence was seen in the control animals (Fig. 1h and j). Majority of HTG-positive

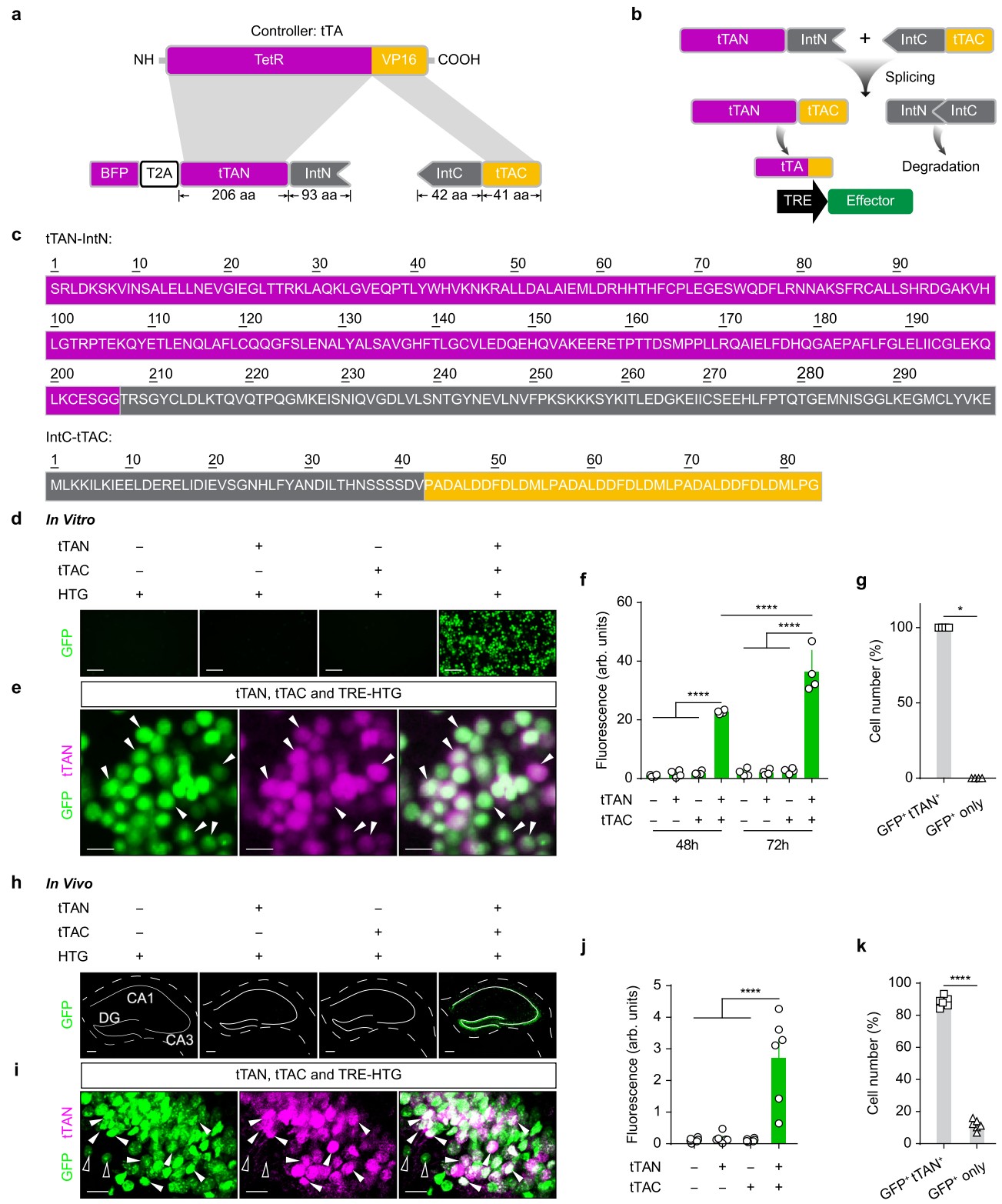

cells co-expressed endogenous fluorescence tag of tTAN (Fig. 1i and k). Together, these results suggest that intein-split tTA/rtTA fragments are capable of forming specific reconstitution in an intersectional manner.

**The function of IBIST-controlled optogenetic opsins and Ca²⁺ indicators.** Combining intersectional approaches with optogenetic manipulations is instrumental to delineate neural circuit

functions[9]. Although many intersectional labeling strategies exist for fluorophore labeling, it remains to be tested whether these strategies are applicable for optogenetic opsins as high expression of opsin is required for optogenetic manipulations in functional studies. We sought to test this for the IBIST tools by injecting the hippocampus with AAVs expressing TRE-oChiEF-mCherry[29] and CaMKIIα-dependent tTAN and tTAC, respectively (Fig. 2a). As expected, potent mCherry fluorescence were only seen in

**Fig. 1 The design of IBIST and the in vitro and in vivo validation. a** Scheme depicting construct design of intein-split tTA. **b** Diagram depicting the reconstitution of tTA by intein-based protein splicing. **c** Amino acid sequence of intein-fused tTA fragments. **d**, Examples of green fluorescence in HEK293T cells 72 h after transfections of reporter plasmid (TRE-HTG) and intein-split tTA plasmids with EF1α promoter (tTAN, tTAC, both or neither). **e** Examples showing the fluorescence of HTG reporter (shown in green) and tTAN (BFP tagged, shown in magenta) in HEK293T cells with all three plasmids transfected. Arrows: co-labeled cells. **f** Summary of green fluorescence intensity (arbitrary units, arb. units) of HEK293T cells (**d**) at 48 h and 72 h post-transfection ($n = 4$ field of view [FOV] in each group). One-way ANOVA: $F_{(7,24)} = 95.73$, ****$P = 5.67 \times 10^{-16}$. Tukey's multiple comparisons test: tTAC+tTAN group at 72 h is significantly higher than that at 48 h post transfection, ****$P = 6.17 \times 10^{-6}$; both are significantly higher than other groups at the same time post-transfection (****$P < 0.0001$ for all). **g** Percentage summary of cells labeled by HTG and tTAN in **e** ($100 \pm 0\%$ co-labeled vs. $0 \pm 0\%$ reporter only; Mann-Whitney U test, *$P = 0.029$, $n = 4$ FOV). **h** Examples showing reporter fluorescence (AAV-TRE-HTG) after its injection into hippocampus and co-injections of AAVs (CaMKIIα promoter) for tTAN, tTAC, both or neither, respectively. **i** Examples showing fluorescence of tTAN (shown in magenta) and HTG reporter (shown in green) in hippocampal cells. Open arrows: reporter only. Filled arrows: co-labeled. **j** Summary of green fluorescence intensities in hippocampus (Each group, $n = 6$ FOV, $N = 3$ animals). One-way ANOVA between groups: $F_{(3,20)} = 20.57$, $P = 2.52 \times 10^{-6}$. Turkey's multiple comparisons: TRE-HTG + tTAC+tTAN group vs. others, ****$P < 0.0001$. **k**, Percentage summary of cells labeled by reporter and tTAN ($88.4 \pm 1.3\%$ co-labeled vs. $11.6 \pm 1.3\%$ reporter only; paired t-test, ****$P = 7.95 \times 10^{-7}$, $n = 6$ FOV, $N = 3$ animals). Scale bars: 200 μm (**h**), 20 μm (**d**, **i**), 5 μm (**e**). Data summary: mean ± SEM. Statistical tests: two sided. Source data are provided as a Source Data file.

animals with all three AAVs injected compared to control animals with AAV-tTAN or AAV-tTAC omitted (Fig. 2b and c). To further validate the faithfulness of reporter expression, we carried out immunostaining to amplify the endogenous fluorescence. We found that nearly all TRE-oChIEF positive cells were also co-labeled by tTAN tag (Supplementary Fig. 2a and b).

Using patch-clamp recording with optogenetic stimuli in acute brain slice (see methods), we found that the blue LED light robustly evoked spikes in hippocampal neurons from the AAV infected brain slices (200 ms constant light stimuli, Fig. 2d and e; 2 ms light pulses at 25 Hz, Supplementary Fig. 2c). These light-evoked spikes are absent in TRE-oChIEF negative cells (Supplementary Fig. 2d–f). In control animals injected with AAVs of tTAN or tTAC omitted, no spikes were evoked by the blue LED light (Fig. 2d and e). We continued to record some of those cells with light stimuli at the maximal power of the LED device (17 mW at the 60X objective), and still observed no spikes in hippocampal cells from control animals (Fig. 2e).

The viral vectors serve as indispensable tools in primate research as transgenic primate animals are very limited. Thus, we further tested whether the same viral vectors were able to reconstitute tTA in the primate brain. We injected same AAVs into primate visual cortex and performed patch-clamp recording with optogenetic stimuli (Fig. 2f). The fluorescence tag of effector (shown in green) was only seen in brain slices with all three AAV vectors injected but not in brain slices with control injections (Fig. 2g and h). Accordingly, low-intensity LED light (3 mW) successfully elicited spikes in all fluorescent cells but not in cells from slices with control injections (Fig. 2i and j).

Next, we validated the optogenetic inhibition by Halorhodopsin (NpHR)[30] under the control of reconstituted tTA (Fig. 3a–c). In the acute brain slices from mice injected with all three AAV vectors, the yellow light greatly inhibited neuronal spikes evoked by current injections (Fig. 3d and e). Such magnitude of light-induced reductions in neuronal spikes were absent in control slices from animals with either AAV-tTAN or AAV-tTAC omitted (Fig. 3d and e). Taken together, these results suggest that the reconstituted tTA is capable of driving sufficient and specific expression of optogenetic opsin for neuronal manipulations in both mice and primates.

Monitoring specific population of neurons defined by intersectional conditions is extremely useful to dissect neuronal functions but also requires high expression of $Ca^{2+}$ indicators in vivo[31,32]. To test whether the IBIST is capable of driving sufficient and specific expression of $Ca^{2+}$ indicators such as GCaMP6s[33], we injected into the hippocampus with AAVs of CaMKIIα-tTAN, CaMKIIα-tTAC and TRE-GCaMP6s. After implanting with optical fiber above the hippocampus

(Supplementary Fig. 3a), we performed photometric recording of $Ca^{2+}$ activity in the hippocampus while the animal was exploring in the context (Supplementary Fig. 4a). When the animal received foot shocks, we observed that strong $Ca^{2+}$ activity was evoked by the foot shocks whereas no detectable $Ca^{2+}$ signal was recorded in control animals with either AAV-CaMKIIα-tTAN or AAV-CaMKIIα-tTAC omitted (Supplementary Fig. 4b–e). Interestingly, we found that the shock-evoked $Ca^{2+}$ activity was further enhanced in the second shock session, indicating that these hippocampal neurons became sensitized to the shock stimuli. Importantly, the $Ca^{2+}$ signal was still not detectable in the control animals (Supplementary Fig. 4e). Taken together, these results confirmed that IBIST is able to drive specific and sufficient effector expression for optogenetic opsins and $Ca^{2+}$ indicators in vivo.

**IBIST-based cell types defined by genetic marker and neural connectivity.** The genetic marker, neural connectivity and neuronal location constitute the major features to define the cell type. While neuronal locations are conveniently controlled by stereotaxic injections of viral vectors, we sought to test the compatibility of IBISIT-based tools when defining the other two types of features in various means. We first tested whether IBIST tools were able to combine features of anterograde labeling and promoter-based viral labeling. The hippocampal CA3 cells send collaterals to both pyramidal cells and interneurons in the CA1 area[34–36]. We designed a strategy to combine AAV-serotype-1-mediated anterograde transsynaptic tracing[37] with pyramidal-cell-specific fluorophore labeling in ventral CA1 (vCA1), aiming to specifically label pyramidal cells with direct inputs from dorsal CA3 (dCA3). We injected AAV1-EF1α-tTAN into the dorsal hippocampus and AAV-CaMKIIα-tTAC and reporter AAV-TRE-HTG into vCA1 (Supplementary Fig. 5a). Although AAV1-EF1α-tTAN travelled in both retrograde and anterograde directions and in turn labeled cells in DG, CA1 and subiculum, we observed specific labelling of CA3-targeted CA1 pyramidal cells owing to the local infection of AAV-CaMKIIα-tTAC in vCA1 (Supplementary Fig. 5b and c). As comparison, no GFP expression was detected in CA1 from the control animals. We then tested whether IBIST tools were able to combine features of anterograde labeling and Cre-dependent labeling in transgenic mice. Using AAV1-Flpo as an anterograde tracer, we specifically labeled *Parvalbumin* expressing (PV+) cells with direct inputs from dCA3 and combined injections of AAV-FRT-BFP-tTAN, AAV-DIO-tTAC and AAV-TRE-GCaMP6s in PV-ires-Cre mice (Supplementary Fig. 5d). As a result, we observed a specific labeling of CA3-targeted PV+ cells in vCA1 (Supplementary Fig. 5e and f). We further tested whether IBIST tools are able to combine features of retrograde rabies tracing and

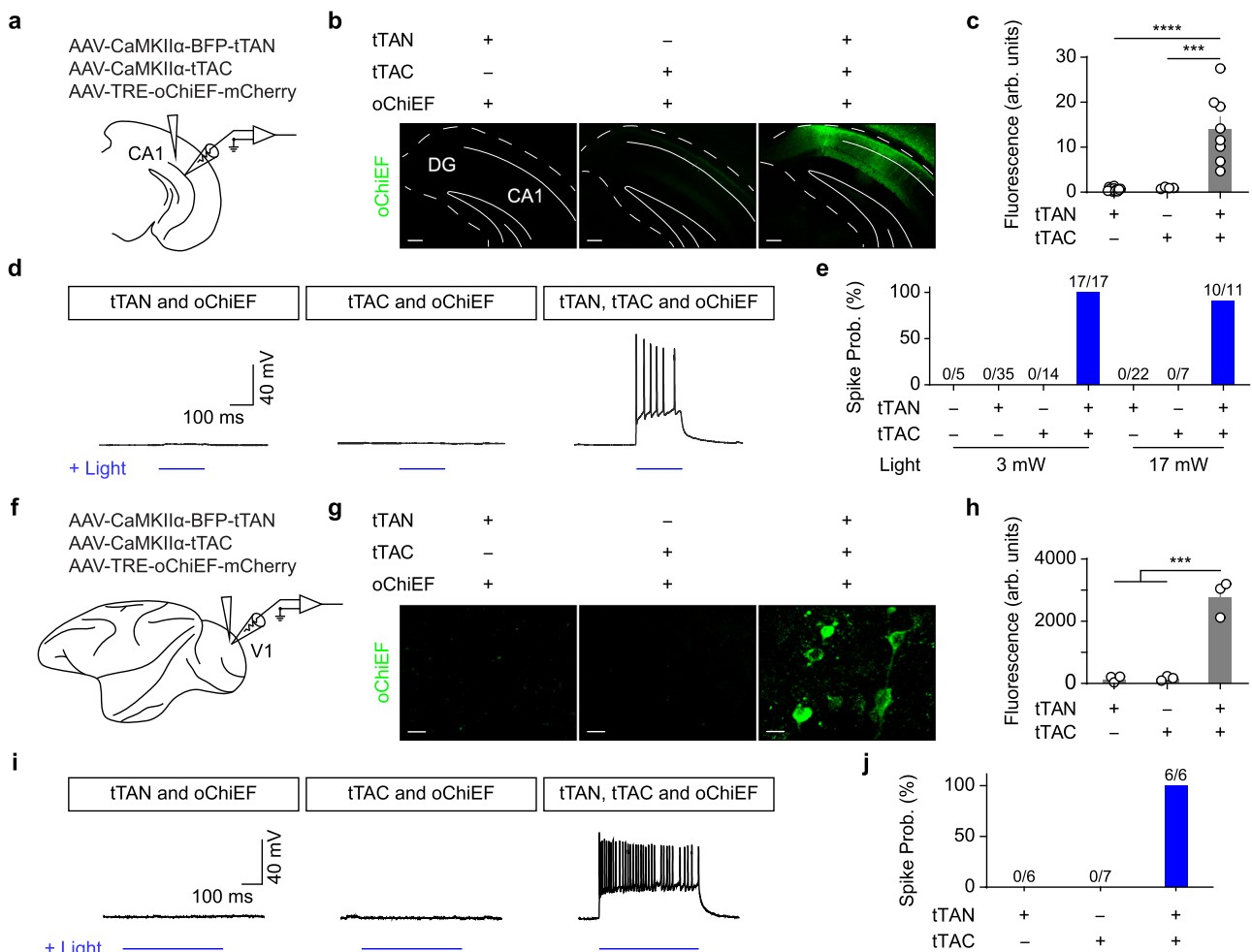

**Fig. 2 The IBIST-based optogenetic excitation. a** Scheme depicting AAV injections into hippocampus. **b**, Examples showing mCherry fluorescence (shown in green) in hippocampus injected with AAV-TRE-oChiEF and AAVs (CaMKIIα promoter) of tTAN, tTAC or both, respectively. Scale bars: 200 μm. **c** Summary of fluorescent intensity in **b**. One-way ANOVA between groups ($F_{(2,21)} = 25.26$, ****$P = 2.58 \times 10^{-6}$). Turkey's multiple comparisons revealed that the fluorescence is significantly higher in tTAN+tTAC group (tTAN vs. tTAN+tTAC, ****$P = 3.07 \times 10^{-6}$; tTAC vs. tTAN+tTAC, ***$P = 0.0002$). tTAN, $N = 3$ animals; tTAC, $N = 1$ animal; tTAN+tTAC, $N = 2$ animals. **d** Examples showing whole-cell current-clamp recording from hippocampal cells in brain slices with blue LED light stimulation (blue line). All animals were injected with AAV-TRE-oChiEF and co-injected with AAVs (CaMKIIα promoter) of either tTAN, tTAC or both. **e** Summary of spike probabilities (prob.) in hippocampal cells evoked by whole-field blue LED light (3 mW or 17 mW) from animals injected with AAV-TRE-oChiEF and other CaMKIIα-specific AAVs: none, $N = 1$ animal; tTAN, $N = 6$ animals; tTAC, $N = 2$ animals; tTAN+tTAC, $N = 4$ animals. The recordings were from local neurons at the injected site. **f** Scheme depicting AAV injections into V1 of *Macaca fascicularis*. **g**, Examples showing mCherry fluorescence (shown in green) in V1 after injected with AAV-TRE-oChiEF-mCherry and CaMKIIα-specific AAVs of tTAN, tTAC or both. Scale bars: 10 μm. **h**, Summary of fluorescent intensity in **g**. Each group consists of brain slices obtained from corresponding injection site ($n = 3$ FOV/group). One-way ANOVA between groups ($F_{(2,6)} = 58.19$, ***$P = 0.0001$) and Turkey's multiple comparisons revealed that the fluorescence is significantly higher in tTAN+tTAC group (***$P < 0.001$ for all). **i** Examples showing whole-cell current-clamp recording in brain slices with blue LED light stimulation (blue line). All groups were injected with AAV-TRE-oChiEF and co-injected with CaMKIIα-specific AAVs of either tTAN and tTAC, or both. **j** Summary of spike probabilities evoked by whole-field blue LED light (3 mW) from animals injected with AAV-TRE-oChiEF and other AAVs. Data summary: mean ± SEM. Statistical tests: two sided. Source data are provided as a Source Data file.

promoter-based viral labeling. We engineered tTAN-expressing rabies vectors and specifically labeled vCA1-projecting dCA3 neurons with TRE reporter (Supplementary Fig. 5g–i). We continued to test whether IBIST tools are able to combine features of both anterograde and retrograde connections. The nucleus of reunien (RE) is known to be a critical relay to transfer information from the medial prefrontal cortex (mPFC) to the hippocampus[38]. By local injections of anterograde AAV1-EF1α-tTAN into mPFC, retrograde AAVretro-CaMKIIα-tTAC into vCA1 and AAV-TRE-oChiEF-mCherry into RE respectively, we specifically labeled RE cells with direct inputs from mPFC and direct outputs to vCA1 (Supplementary Fig. 5j and k). No cells were labeled if either tTAN or tTAC component was omitted

from injections (Supplementary Fig. 5k). These results demonstrated that IBIST-based tools are capable of defining cell types with the genetic marker and neural connectivity by various means using viral vectors and transgenic animals.

To validate whether these dual-feature cell types have sufficient gene expression for effectors such as $Ca^{2+}$ indicators and optogenetic opsins, we defined vCA1 cells by the genetic marker and their connections with dCA3. To specifically image vCA1 excitatory neurons with direct inputs from dCA3, we injected AAV1-EF1α-Flpo into dCA3 and injected into vCA1 with AAV-FRT-BFP-tTAN, AAV-CaMKIIα-tTAC and effector AAV-TRE-GCaMP6s (Fig. 4a±c). We performed photometric recording from those excitatory neurons in vCA1 while animals underwent open field test (Fig. 4d, e and

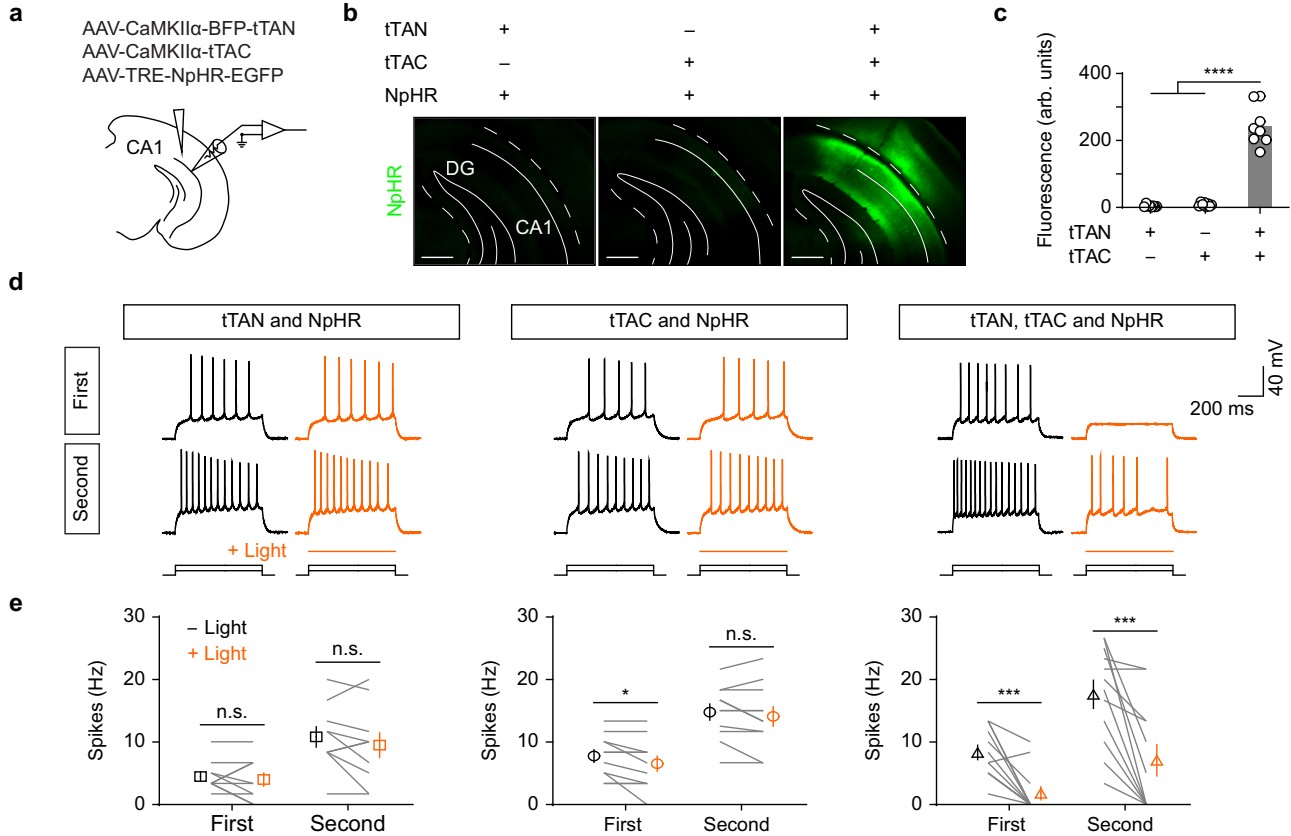

**Fig. 3 The IBIST-based optogenetic inhibition. a** Scheme depicting AAV injections into hippocampus. **b** Examples showing GFP fluorescence in hippocampus after injected with AAV-TRE-NpHR and AAVs (CaMKIIα promoter) of tTAN, tTAC or both, respectively. Scale bars: 500 μm. **c**, Summary of fluorescent intensity in **b**. One-way ANOVA revealed significant fluorescence differences between groups ($F_{(2,21)} = 123.1$, ****$P = 2.52 \times 10^{-12}$) and Turkey's multiple comparisons revealed that the fluorescence is significantly higher in tTAN+tTAC group than in others (****$P < 0.0001$ for all). $N = 2$ animals for each group. **d** Examples showing whole-cell current-clamp recording from hippocampal cells in brain slices upon current injections with 40 pA steps (first and second sweeps with prominent spikes; yellow with light; black without light). **e** Frequency of spikes in hippocampal cells in the absence (black) or presence (yellow) of 589 nm laser light (18 mW) from animals injected with AAV-TRE-NpHR and other CaMKIIα AAVs. The tTAN+tTAC animals showed a significant decrease in frequency of spikes after light stimulation (Wilcoxon matched-pairs signed rank test; first sweep, $8.3 \pm 1.1$ Hz vs. $1.8 \pm 1.0$ Hz with light; ***$P = 0.001$; second sweep, $17.6 \pm 2.3$ Hz vs. $7.1 \pm 2.5$ Hz, ***$P = 0.001$; $n = 12$ cells, $N = 2$ animals). The tTAN animals, paired $t$-test; first sweep, $4.5 \pm 0.7$ Hz vs. $4.0 \pm 1.1$ Hz with light, $P = 0.5414$; second sweep, $10.8 \pm 1.6$ Hz vs. $9.5 \pm 2.0$ Hz with light; $P = 0.21$; $n = 10$ cells, $N = 2$ animals. The tTAC animals, paired $t$-test; first sweep, $7.7 \pm 1.0$ Hz vs. $6.5 \pm 1.2$ Hz with light, *$P = 0.04$; second sweep, $14.8 \pm 1.3$ Hz vs. $14.1 \pm 1.5$ Hz with light, $P = 0.25$; $n = 11$ cells, $N = 3$ animals. Data summary: mean ± SEM. Statistical analysis: two sided. Source data are provided as a Source Data file.

Supplementary Fig. 3b). We observed significant increase in $Ca^{2+}$ signals when animals were entering the center of the open field, which were not seen in control animals (Fig. 4f and g). In contrast, we did not observe significant increase in $Ca^{2+}$ signals when animals exited the center. We found that both entry and exit showed similar speed dynamics (Fig. 4f). These results indicated that these CA3-targeted vCA1 cells exhibit anxiogenic signals. We also observed significant $Ca^{2+}$ signals in these CA1 cells when animals received foot shocks (Fig. 4h–j). Next, we adopted similar IBIST-based strategy to enable specific ChR2 expression in pyramidal cells with dCA3 inputs (Supplementary Fig. 6a–d). The blue light successfully evoked spikes in vCA1 neurons receiving inputs from dCA3 (Supplementary Fig. 6e and f). Finally, we utilized different sets of viral vectors to enable specific NpHR expression in *Somatostatin* expressing (SOM+) cells with dCA3 inputs in SOM-ires-Cre mice (Supplementary Fig. 6g and h). As expected, the yellow light effectively cracked down the neuronal spikes (Supplementary Fig. 6i and j). These results confirmed that the IBIST tools enabled dual-feature cells with specific and sufficient expression of $Ca^{2+}$ indicators and optogenetic opsins.

In summary, both tTAN and tTAC are compatible with viral vectors for anterograde and retrograde circuit tracing and controllable by recombinase, such as Cre and Flpo. Therefore, these IBIST tools are able to define cell types with two types of features by various means, achieve cell-type specific fluorophore expression and suffice for $Ca^{2+}$ imaging and optogenetic manipulation.

**IBIST-based $Ca^{2+}$ imaging for cells with multiple projections**. To exploit the application of IBIST tools to label cells with multiple projections, we turned to ventral hippocampus where neuronal functions diversify by their downstream targets[39–43]. For example, the NAc projector contributed to drug-induced place preference[39] and social memory[40]. The amygdala projector contributed to fear conditioning[41] and the lateral septum (LS) projector contributed to feeding[44]. Interestingly, mPFC and amygdala projection cells exhibited distinct functions in fear extinction compared to single-projection cells[45]. In line with this, we found that mPFC-amygdala-projectors exhibited different biophysical properties from the single projectors (Supplementary Fig. 7 and Supplementary Table 1). Recent studies showed that different subgroups of vCA1 neurons connected with distinct patterns of downstream areas[46] and exhibited distinct physiological functions[43]. Thus, it is important

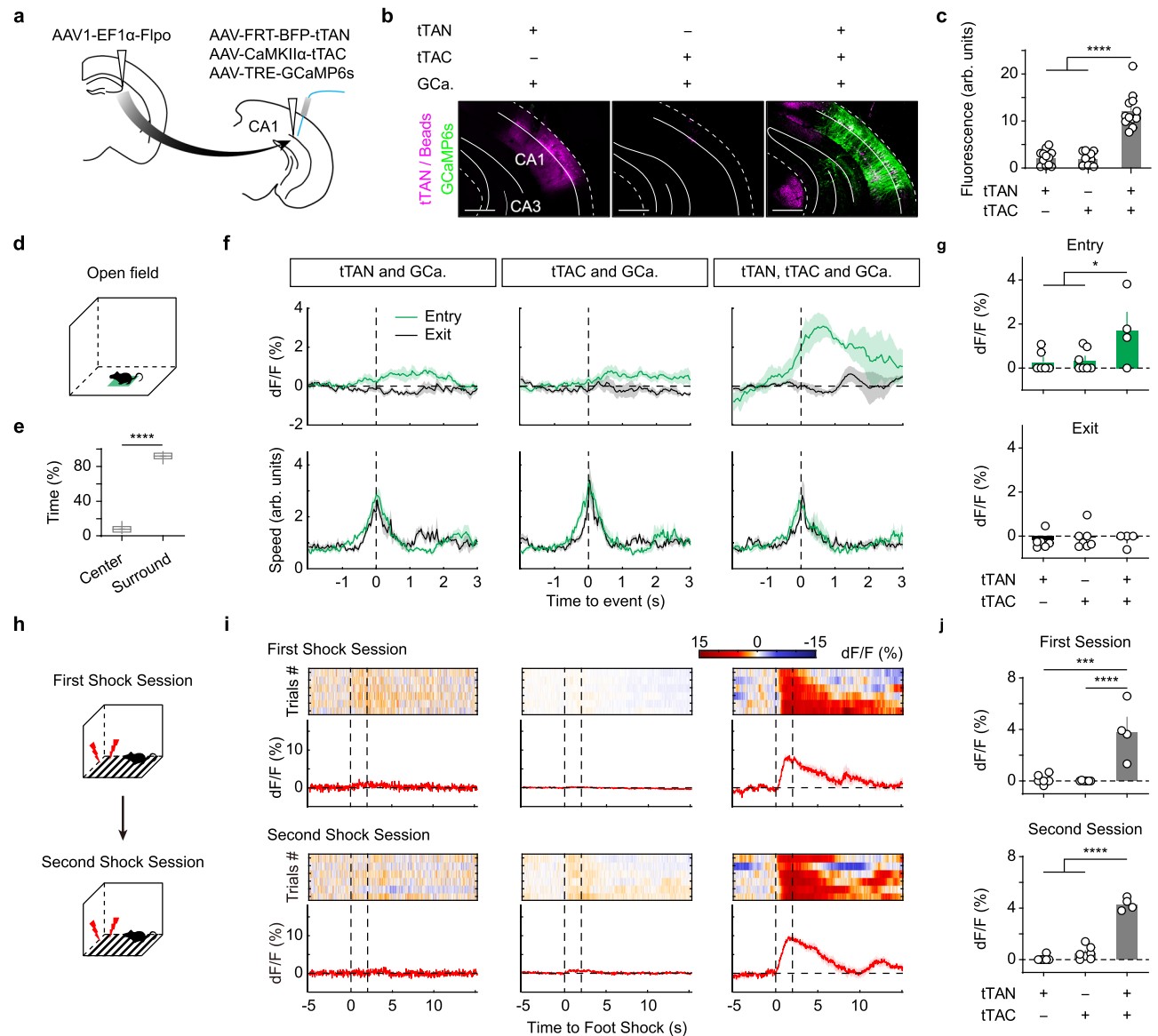

**Fig. 4 The IBIST-based Ca²⁺ recording for cells defined by genetic marker and neural connectivity. a** Scheme illustrating AAV injections to define cells by two features: anterograde transsynaptic tracing and CaMKIIα-specific fluorophore labeling. **b** Examples showing fluorescence of the ventral hippocampus infected by AAVs of Flpo, tTAN (magenta), tTAC and TRE-GCaMP6s (GCa., green) and co-injected with blue beads (magenta). Scale bars: 500 μm. **c**, Summary of fluorescent intensity of GCaMP6s in **b**. One-way ANOVA between groups: $F_{(2,33)} = 66.19$, ****$P = 2.82 \times 10^{-12}$. Turkey's multiple comparisons: tTAN vs. tTAN+tTAC, ****$P = 4.91 \times 10^{-11}$; tTAC vs. tTAN+tTAC, ****$P = 3.88 \times 10^{-11}$. $N = 3$ animals/group. **d** Scheme illustrating the open field test. **e** Box plot of the percentage time in the center zone and surround. Center vs. surround, $8.2 \pm 1.0\%$ vs. $91.8 \pm 1.0\%$, paired $t$-test, ****$P = 2.08 \times 10^{-18}$; $N = 17$ animals. **f** Examples of averaged traces of Ca²⁺ signals and animal speeds upon center entry (green) and exit (black) in the open field test. **g** Summary of normalized Ca²⁺ responses upon center entry and exit. One-way ANOVA between groups (tTAN, $0.3 \pm 0.2\%$, $N = 6$ animals; tTAC, $0.4 \pm 0.2\%$, $N = 7$ animals; tTAN+tTAC, $1.7 \pm 0.8\%$, $N = 4$ animals; $F_{(2,14)} = 4.275$, *$P = 0.0356$; Turkey's multiple comparisons, *$P < 0.05$). **h** Scheme illustrating the foot shock test. **i** Examples of heatmaps and averaged traces of Ca²⁺ signals upon foot shocks (between dash lines,). **j**, Summary of Ca²⁺ responses. One-way ANOVA between groups in first session (tTAN, $0.1 \pm 0.2\%$, $N = 6$ animals; tTAC, $0.01 \pm 0.01\%$, $N = 7$ animals; tTAN+tTAC, $3.9 \pm 1.1\%$, $N = 4$ animals; $F_{(2,14)} = 21.17$, ****$P = 5.7 \times 10^{-5}$; Turkey's multiple comparisons, ***$P < 0.001$) and second session (tTAN, $0.1 \pm 0.1\%$, $N = 6$ animals; tTAC, $0.5 \pm 0.2\%$, $N = 7$ animals; tTAN+tTAC, $4.3 \pm 0.2\%$, $N = 4$ animals; $F_{(2,14)} = 141.3$, ****$P = 5.23 \times 10^{-10}$; Turkey's multiple comparisons, ****$P < 0.0001$). Data summary: mean ± SEM. Statistical analysis: two sided. Box plots: whiskers (min/max), middle line (median), plus (mean), box (25/75 percentile). Source data are provided as a Source Data file.

and interesting to investigate how different subgroups of vCA1 projection cells respond to external stimuli with distinct emotional valences. To address this question, we used IBIST tools to label vCA1 cells with triple features defined by two projection targets and pyramidal cell-specific CaMKIIα promoter. We injected AAVretro-CaMKIIα-tTAN and AAVretro-CaMKIIα-tTAC into one or two vCA1 downstream targets including amygdala, nucleus accumbens (NAc) and mPFC respectively, and injected AAV-TRE-GCaMP6s into vCA1 (Fig. 5a). We found that the double-projection cells were successfully labeled, which were not seen in control experiments with either AAVretro-CaMKIIα-tTAN or AAVretro-CaMKIIα-tTAC omitted (Fig. 5c). Using photometric recording from the vCA1 when animals underwent conditioned place preference (CPP) training and two foot-shock sessions

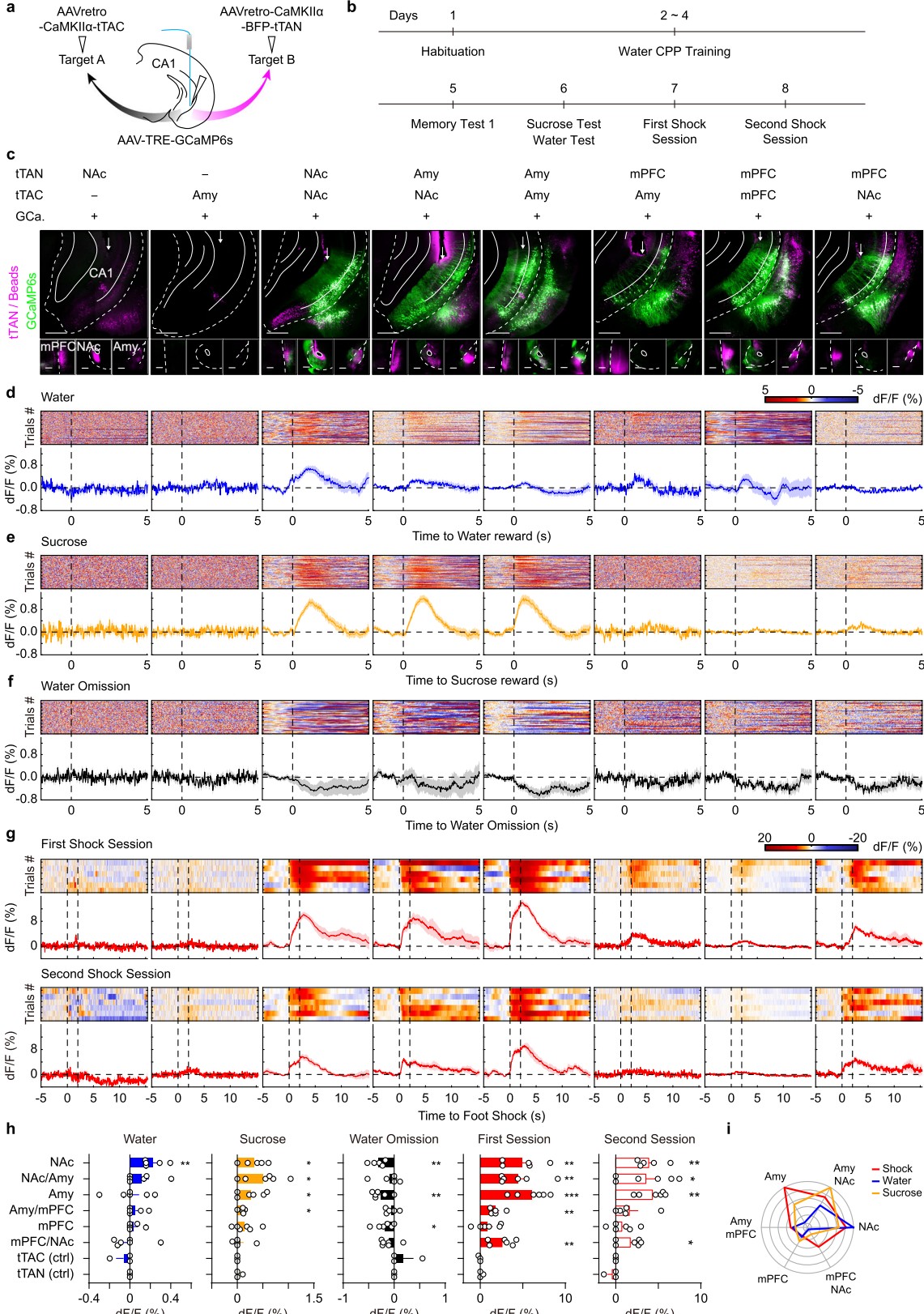

(Fig. 5b, Supplementary Fig. 3c, Supplementary Fig. 8), we observed that distinct Ca$^{2+}$ activity was evoked in different types of vCA1 projection cells by the water reward, sucrose reward and foot shock (Fig. 5d–g). No detectable Ca$^{2+}$ signals were recorded in the control animals (Fig. 5d–g). For the sucrose reward, the NAc-projecting and NAc-Amy-projecting vCA1 cells responded strongly and all types of vCA1 cells with mPFC connections responded weakly, if not none (Fig. 5h and i). Notably, water reward elicited specific activations of NAc-projecting and NAc-Amy-projecting vCA1 cells. Interestingly, the omission of water

**Fig. 5 The IBIST-based Ca$^{2+}$ recording for cells defined by multiple projections. a** Scheme illustrating AAV injections to define cells by three features: dual retrograde tracings and pyramidal-cell-specific (CaMKIIα promoter) Ca$^{2+}$ recording. **b** Schematic showing the behavioral design of water CPP and shock sessions. **c** Examples of fluorescence images in vCA1 (top, injected with AAV-TRE-GCaMP6s, shown in green) and downstream regions (bottom). AAVretro-tTAN (shown in magenta) and AAVretro-tTAC were injected into two downstream regions, co-injected into one region, or omitted. All the AAVs were co-injected with blue beads (shown in magenta). The arrows indicate optical fiber tracks. mPFC: medial prefrontal cortex. NAc: nucleus accumbens. Amy: amygdala. Scale bars: 500 μm. **d–g,** Examples of heatmaps and averaged traces of Ca$^{2+}$ signals upon water reward (**d**, blue), sucrose reward (**e**, yellow), water omission (**f**, black) and foot shock (**g**, red). **h** Summary of Ca$^{2+}$ responses in vCA1 depending on targeted areas. The significance for Ca$^{2+}$ response in each group is determined by one sample $t$-test (H0: μ = 0), *$P$ < 0.05, **$P$ < 0.01, ***$P$ < 0.001 . **i,** Polar summary plot of Ca$^{2+}$ responses to foot shocks and water and sucrose rewards normalized by the max response in each type of emotional stimuli. Data summary: mean ± SEM, details in Supplementary Table 2. Statistical analysis: two sided, details in Supplementary Table 3. Source data are provided as a Source Data file.

reward elicited noticeable inhibitory signals in NAc-projecting and Amy-projecting vCA1 cells (Fig. 5h and i). For aversive foot shocks, vCA1 cells were activated in a target-dependent manner (Fig. 5h and i). The foot shock-evoked Ca$^{2+}$ activity was strong in Amy-projecting and NAc-projecting cells, weak in mPFC-projecting cells and moderate in the double-projecting cells (see detailed summary and statistics in Supplementary Tables 2 and 3).

Among the double-projecting vCA1 cells, mPFC-NAc-projectors preferentially responded to foot shocks but not sucrose rewards, whereas NAc-Amy-projectors strongly responded to both (Fig. 5i). These results promoted us to examine their causal roles in the emotion processing. To address this, we employed the same strategies of AAV injections and selectively expressed optogenetic opsins in mPFC-NAc-projecting and NAc-Amy-projecting vCA1 cells, respectively. Using a real-time place preference paradigm, we performed bidirectional manipulations of those vCA1 cells when animals were exploring in the context. We found that activating mPFC-NAc-projectors led to a negative preference for context paired with light stimulation, while inhibiting mPFC-NAc-projectors led to a positive preference for stimulated context (Supplementary Fig. 3d and e and Supplementary Fig. 9). In contrast, there were no significant behavioral effects by manipulating NAc-Amy-projectors (Supplementary Fig. 9). These optogenetic results, together with photometric results, suggest that mPFC-NAc-projectors preferentially mediate negative emotion processing.

Finally, we tested whether the IBITS tools could label specific types of hippocampal cells based on more downstream targets. We injected AAV-TRE-GCaMP6s into vCA1 and AAVs (AAVretro) of EF1α-FRT-tTAN, EF1α-Flpo, EF1α-DIO-tTAC, CaMKIIα-Cre into NAc, mPFC, LS and amygdala, respectively. As expected, we observed prominent fluorescence of GCaMP6s in vCA1, which was absent in controls where either tTAN or tTAC was omitted from injections (Fig. 6a and b, Supplementary Fig. 3f). Accordingly, we recorded robust Ca$^{2+}$ signals from vCA1 upon foot shock (Fig. 6c and d). Interestingly, these Ca$^{2+}$ signals exhibited adaption over trials (first session, first 3 trials vs. last 3 trials, 2.1 ± 0.7% vs. 1.2 ± 0.4%; paired $t$-test, $P$ = 0.0349). Therefore, the IBIST-based Ca$^{2+}$ imaging is applicable for cell types defined by five features including one promoter and four different projection targets. Taken together, the selective recording of Ca$^{2+}$ activity in various types of projection cells in vCA1 revealed distinct activity profiles in response to aversive and appetitive stimuli, highlighting the importance to define cell types based on multiple projection targets.

## Discussion
Our work demonstrated that the IBIST can be efficiently reconstituted in various ways depending on promoters, brain regions, neuronal connections and site-specific recombinases. The IBIST system allows for more intersectional features than prior tools by multiplexing the number of fragmented tTA (Supplementary Fig. 10) and enables technological features such as ease to design

the IBIST construct, reconstitution of tTA at the protein level, common TRE-effectors for various experimental strategies, reversible control by doxycycline, compatibility with Cre and Flp transgenic animals and compatibility with rabies-based transsynaptic tracing.

While intersectionally labeling neurons by fluorophores is widely used, it is still challenging to achieve the specificity and sufficiency for optogenetic manipulations and Ca$^{2+}$ imaging in vivo. Pioneering works developed intersectional effectors for opsins and Ca$^{2+}$ indicators[12], but new combinations of controllers would still require sophisticated construct design, which may suffer from insufficient gene expression in some conditions. The IBIST tools could help to resolve these concerns by intersectional synthesis of controller (tTA) which does not need that high expression level as the effector does. Accordingly, it is more straightforward to design constructs for IBIST effectors and utilize a large library of ready-to-use common effectors, and more feasible to achieve sufficient gene expression for opsins and Ca$^{2+}$ indicators. In our work, we have validated TRE-based viral effectors of oChIEF (ChR2 variant)[29], NpHR[30], GCaMP6s[33] and HTG (histoneGFP-TVA-Glycoprotein)[27], which can be broadly utilized in various conditions with viral serotypes, promoters and transgenic animals. As the design of splitting tTA is also applicable for splitting rtTA (Supplementary Fig. 1b–f), there could be more adaptable options for intersectional control in various experimental purposes.

The intein mediated splicing technique has long been used before, but the efficiency is varying depending on the intein sequence and the application context. We adopted a so-far best strategy based on Gp41-1 sequences because they have small sizes and possess the most rapid reaction rate among all split inteins examined[18,21]. In the future, mapping split site of Gp41-1 amino acid sequence for more efficient splicing and compatible segmentation of various controllers might bring further improved performance for IBIST tools. We have validated the specificity of IBIST tools by careful evaluation and noticed a small bit of leak by AAV-tTAN or AAV-tTAC when the titer was higher than 10$^{13}$. While care should be taken when using all AAV vectors at high titers, we recommend to simply use AAVs with titers at the level of 10$^{12}$ in order to avoid potential non-specific gene expression. Future work may further ensure the specificity by fusing tTAN and tTAC with destabilizing domains such as ddFKBP and SopE[47–50] which help to clear residual of tTAN or tTAC in the cells accordingly. In our experiences, some AAV-TRE effectors have faint baseline expression of fluorophore tag and hence are not suitable for simple fluorescent tagging. Therefore, the strength of our system lies more in the specific and sufficient gene expression for effectors such as optogenetic opsins and Ca$^{2+}$ indicators. For instance, we have successfully used the IBIST system to record subpopulations of vCA1 neurons targeting different combinations of brain areas (Figs. 5 and 6). Recent studies suggested that the physiological functions of hippocampal projectors depend on their downstream targets[39–43]. We found

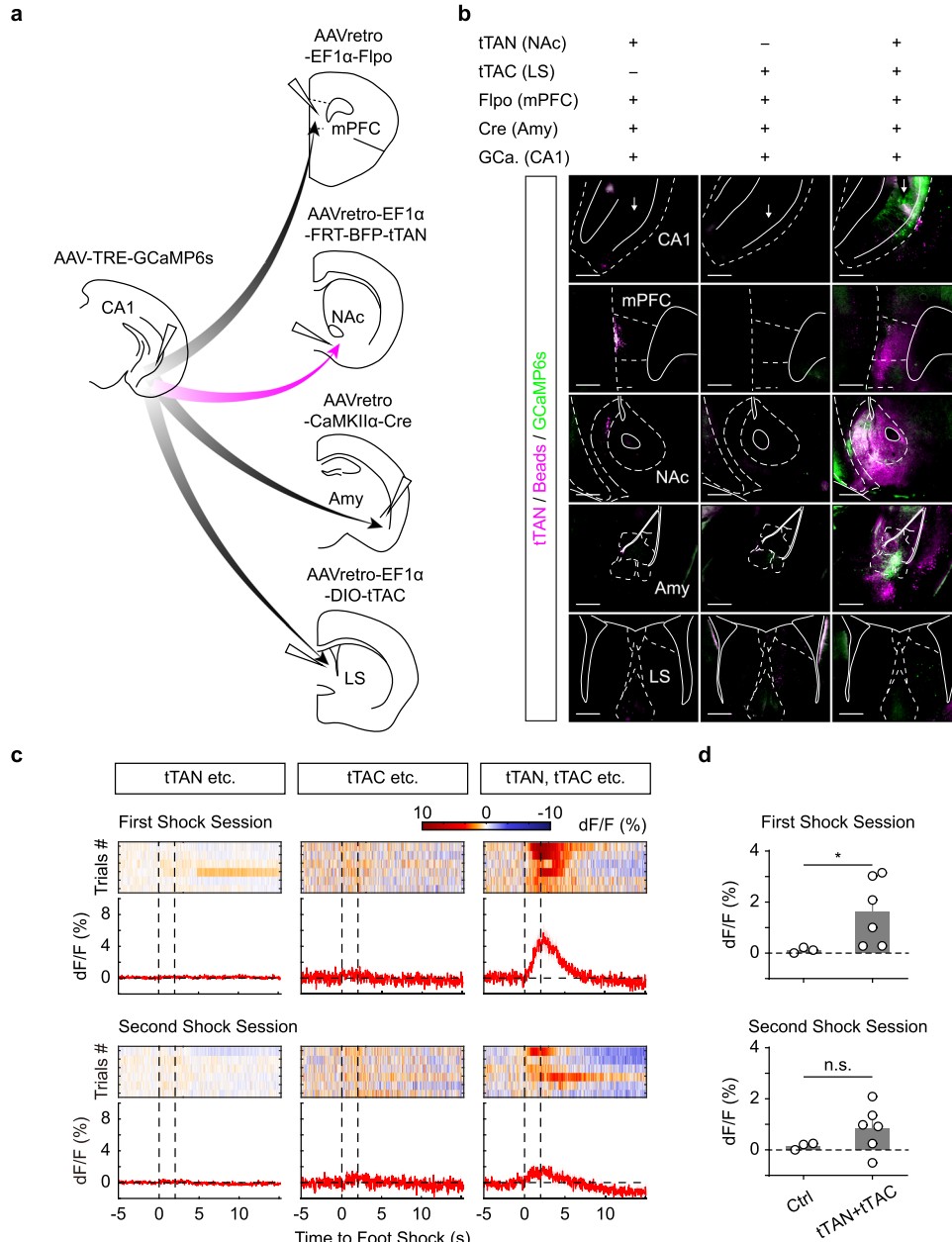

**Fig. 6 Quintuple features-based Ca²⁺ recording. a** Scheme illustrating AAV injections to define cells by five features including quadruple retrograde tracings and pyramidal-cell-specific (CaMKIIα promoter) Ca²⁺ recording. **b** Examples showing fluorescence of GCaMP6s (green), tTAN (magenta) and blue beads (magenta). mPFC: medial prefrontal cortex. NAc: nucleus accumbens. Amy: amygdala. LS: lateral septum. Scale bars: 500 μm. **c** Examples of heatmaps and averaged traces of Ca²⁺ signals upon foot shocks between dash vertical lines. **d** Summary of foot shocks evoked Ca²⁺ responses in tTAN +tTAC ($N = 6$ animals) and control (1 animal for tTAN and 2 animals for tTAC) groups. First session, 1.6 ± 0.5% vs. 0.1 ± 0.06%; Mann-Whitney U test, *$P = 0.0238$; Second session, 0.8 ± 0.4% vs. 0.2 ± 0.1%, Mann-Whitney U test, $P = 0.26$. Data summary: mean ± SEM. Statistical analysis: two sided. Source data are provided as a Source Data file.

that the NAc projectors in vCA1 responded to both sucrose rewards and aversive foot shocks (Fig. 5h and i). This is consistent with the notion that the NAc contributes to both appetitive and fearful motivations[51,52]. Interestingly, the NAc-mPFC-projectors preferentially responded to foot shocks (Fig. 5h and i) and causally regulated real-time place preference (Supplementary Fig. 9). As a comparison, the NAc-Amy-projectors may function as salience detectors by responding to all rewards and aversive shocks (Fig. 5i). Thus, our study using the IBIST system outlines an emerging picture for the projection-pattern-dependent emotional processing in hippocampal neurons (Fig. 5). Future studies with

IBIST tools could further reveal specific physiological functions of more types of multi-area projectors[46].

Our IBIST system is not only compatible with increasing mouse driver lines with Cre and Flp for most major cell types in the brain[5,6,24], but also complementary for the existing intersectional tools such as Cre- and Flp-dependent constructs created by pioneering works[4,12]. Intersectional control with more than five features (Supplementary Fig. 10) is possible if the split tTA fragments are built upon those Cre- and Flp-dependent constructs[12], antero- and retrograde tracing or other promoter and enhancer-dependent constructs[53,54]. Intersectional ON and

OFF is also possible by the IBIST tools if one of the tTA fragments is flanked by palindromic sites recognized by site-specific recombinases (e.g. Cre-loxp site-specific recombination system). As the intein-split strategy could be adapted for other controllers such as Cre[7], it would be possible to devise two orthogonal strategies in the same animal by independent and parallel synthesis of two single controllers in intersectional manners, respectively. Given that the IBIST tools can be entirely implemented in viral vectors including AAV and rabies vectors, our system not only offers experimental conveniences but also holds great promise as intersectional tools for animal species with limited transgenic resources such as primates.

## Methods

**Plasmids**. All vectors in this study were constructed using EasyGeno Assembly Cloning kit (TIANGEN, Beijing) according to the manual, except for the specially noted plasmids. Briefly, the backbone plasmid was linearized and the insert fragments were amplified through normal polymerase chain reactions. All DNA products were separated in agarose gel to confirm sizes and following purified using TIANgel Midi Purification Kit (TIANGEN). The mixture of 0.01 pM of backbone and 0.03 pM of each insert fragments were delivered for transformation in Stable3 competent bacteria after 30 minutes incubation in 50 °C. All the plasmids are verified by sequencing.

The intein sequence for N-terminal part in this study is:
acccgcagcggctactgcctggaacctgaagacccaggtgcagacccccccagggcatgaaggagatcagcaacatccaggtgggcgacctggtgctgagcaacaccggctacaacgaggtgctgaacgtgttccccaagagcaagaagaagagctacaagatcaccctggaggacggcaaggagatcatctgcagcgaggagcacctgttccccacccagaccggcgagatgaacatcagcggcgcctgaaggagggcatgtgcctgtacgtgaaggag.

The intein sequence for C-terminal part in this study is:
atgctgaagaagatcctgaagatcgaggagctggacgagcgcgagctgatcgacatcgaggtgagcggcaaccacctgttctacgccaacgacatcctgacccacaacagcagcagcagcgacgtg.

The plasmids created in this study are available on Addgene (#172119 – 172127). The response plasmids for Tet-On and Tet-Off gene expression all contain Tet-responsive Ptight promoter.

In this study, pAAV-TRE-HTG (Addgene, #27437) was a gift from L. Luo[27]; pAAV-EF1α-DIO-TVA950-T2A-CVS11G was a gift from B. Roska[55]; pcDNA-B19N, pcDNA-B19P, pcDNA-B19L, pcDNA-B19G and pSADdeltaG-F3 (#32634) were gifts from E.M. Callaway[56]; pLenti-CaMKIIα-VChR1-EYFP (#20954) was a gift from K. Deisseroth[57]; pLKO-GFP-IntN-SopE and pLKO-IntC-Flag were gifts from Z.-K. Qian[58]; pmSyn1-EBFP-Cre (#51507) was a gift from H. Zeng[6]; pAAV-TRE-fDIO-GFP-IRES-tTA (#118026) was a gift from M. Luo[59].

### Virus preparations

*Rabies vectors*. The SADΔG rabies virus was similarly generated as before[41]. The G-deleted rabies virus expressing rtTAN-GFP (RV-rtTAN) was recovered by transfecting B7GG cells (kindly provided by Ed Callaway, Salk Institute) with pcDNA-B19N, pcDNA-B19P, pcDNA-B19L, pcDNA-B19G and pRabiesΔG-rtTAN-GFP in a humidified atmosphere of 5% $CO_2$ and 95% air at 35 °C. The recovered RV-rtTAN was amplified in B7GG cells, the supernatant was used to infect HEK293T cells. The titer of the RV-rtTAN viral supernatant was in the range of $10^3$–$10^4$ infectious units/mL.

The G-deleted rabies virus expressing tTAN-BFP (RV-tTAN) was recovered similarly. For in vivo injection, RV-tTAN containing supernatant was concentrated by ~100,000 × *g* centrifuge for 4 h. The titer was in the range $10^6$–$10^7$ infectious units/mL.

*AAV vectors*. AAV vectors were typically generated with serotype 8 and 9, unless otherwise noted in the text. In this study, the effector AAVs for Tet-On and Tet-Off gene expression all contain Tet-responsive Ptight promoter.

Most AAV vectors were produced by the gene editing core facility at the Institute of Neuroscience: AAV2/9-CaMKIIα-BFP-tTAN ($7.2 \times 10^{13}$), AAV2/9-CaMKIIα-tTAC ($4.2 \times 10^{13}$), AAV2/1-EF1α-BFP-tTAN ($4.3 \times 10^{13}$), AAV2/1-EF1α-Flpo ($8 \times 10^{12}$), AAV2/8-EF1α-FRT-BFP-tTAN ($8.9 \times 10^{13}$), AAV2/8-EF1α-DIO-tTAC ($1.4 \times 10^{14}$), AAVretro-CaMKIIα-BFP-tTAN ($1.8 \times 10^{13}$), AAVretro-CaMKIIα-tTAC ($6.8 \times 10^{12}$), AAV2/9-TRE-HTG ($1.8 \times 10^{13}$), AAVretro-EF1α-Flpo ($7.9 \times 10^{12}$), AAVretro-EF1α-FRT-BFP-tTAN ($1.3 \times 10^{13}$), AAVretro-EF1α-DIO-tTAC ($7.5 \times 10^{12}$), AAVretro-CaMKIIα-cre ($2.4 \times 10^{13}$). The TRE-driven AAV effectors were produced with serotype 9 (Taitool Bioscience, Shanghai): AAV2/9-TRE-oChiEF-mCherry ($1.59 \times 10^{13}$), AAV2/9-TRE-GCaMP6s ($2 \times 10^{13}$), AAV2/9-TRE-NpHR-EGFP ($1.5 \times 10^{13}$). The titer of AAVs was typically diluted into the range of $10^{12}$ before in vivo injection. The AAV order information is available at https://github.com/XuChunLab/IBIST.

**Cell culture**. HEK293T (SCSP-502; Cell bank, Shanghai) and BG77 cells (provided by E.M. Callaway) are cultured with 10% (v/v) FBS/DMEM at 5% $CO_2$ and 37 °C.

*Reconstitution of tTA in vitro*. The HEK293T cells were divided into four groups and transfected with plasmids by Lipofectamine 2000 (11668-027, Invitrogen). The four groups (tTAN, tTAC, both or neither) of plasmids for transfection consist of reporter pAAV-TRE-HTG and intein-split tTA plasmids with EF1α promoter. The fluorescence images for HEK293T cells were acquired at 48 h and 72 h after the transfection.

*Reconstitution of rtTA in vitro*. The HEK293T cells were divided into three groups and transfected with plasmids of pAAV-TRE-HTG and pAAV-EF1α-rtTAC by Lipofectamine 2000 (11668-027, Invitrogen). The culture medium was replaced with doxycycline-containing medium (100 ng/ml, MedChemExpress, HY-N0565B) 6 h after the transfection. Eighteen hours later, the culture medium was replaced with RV-rtTAN-containing medium (rabies-containing supernatant from B7GG cells) for one group of cells and incubated for 6 h, and then replaced with fresh medium again (with doxycycline 100 ng/ml). For the control groups, pAAV-EF1α-rtTAC plasmid or RV-rtTAN-containing medium was omitted, respectively. The fluorescence images for HEK293T cells were acquired at 48 h and 72 h after the transfection.

*Quantification for fluorescence and co-labeling in vitro*. The fluorescence images for HEK293T cells were acquired by Olympus microscope (CKX53 with X-CITE LED, 10X objective, same exposure settings for the same type of experiment). One FOV image per culture plate was taken for quantification (green for TRE-HTG and rtTAN-GFP, red for TRE-mCherry and blue for tTAN-BFP). The mean fluorescence intensity in the whole FOV was measured and normalized by the area of FOV in ImageJ. The reporter cells and co-labeled cells were counted in Imaris (v7.4.2, Bitplane).

**Animals**. Animals were housed under a 12 h light/dark cycle with temperature controlled between 21 and 23 °C and humidity-controlled between 40–70%. Animals were provided with food and water *ad libitum* in the Institute of Neuroscience animal facility (mice and cynomolgus monkey). All animal procedures were performed in accordance with institutional guidelines and were approved by the Institutional Animal Care and Use Committee (IACUC) of the Institute of Neuroscience (CAS Center for Excellence in Brain Science and Intelligence Technology), Chinese Academy of Sciences. Wild-type C57BL/6 J (Slac Laboratory Animal, Shanghai), PV-ires-Cre[60] and SOM-ires-Cre[14] mice were used. All the experimental mice used in the study were adult male mice (over 8 weeks). One male cynomolgus monkey (*Macaca fascicularis*, 4.2 kg, 13 years old) was obtained from nonhuman primate facility of Institute of Neuroscience after its full use in reproductive research.

**Stereotaxic injections**. Mice were anaesthetized with isoflurane (induction 5%, maintenance 2%, RWD R510IP, China) and fixed in a stereotactic frame (RWD, China). Before the surgery, local analgesics (Lidocaine, H37022147, Shandong Hualu Pharmaceutical) were administered. A feedback-controlled heating pad (FHC) was used to maintain the body temperature at 35 °C. Glass pipettes (tip diameter 10–20 μm) connected to a Picospritzer III (Parker Hannifin Corporation) were filled with virus solutions and injected at following coordinates (posterior to Bregma, AP; lateral to the midline, LAT; below the brain surface, DV; in mm): mPFC: AP + 1.98, LAT ± 0.6, DV -1.62, angle 10°; Amygdala: AP -1.2, LAT ± 3.0, DV -4.35; NAc: AP + 1.0, LAT ± 1.0, DV -4.2; RE: AP -0.3, LAT ± 1.3, DV -4.5, angle 18°; dCA3: AP -1.5, LAT ± 2.1, DV -1.7; vCA1: AP -3.2, LAT ± 3.4, DV -1.5 or 3.5. To identify the injection site, the virus solution was typically mixed at 1:500 with blue beads (fluorescent polymer microspheres, Thermo Scientific).

*Local injections in hippocampus*. A mixture (300 nL) of AAV2/9-TRE-HTG, AAV2/9-CaMKIIα-tTAC and AAV2/9-CaMKIIα-tTAN-BFP (ratio 1:1:1) were injected into dCA3 (AP -1.5, LAT ± 2.1, DV -1.7). In control groups, one of these three viruses (AAV2/9-TRE-HTG, AAV2/9-CaMKIIα-tTAC and AAV2/9-CaMKIIα-tTAN-BFP) was omitted for injections (ratio 1:1 for two viruses). In negative control group, only AAV2/9-TRE-HTG (300 nL) was injected into dCA3.

For electrophysiological experiments combined with optogenetics, same AAVs were injected into vCA1 (AP -3.2, LAT ± 3.4, DV -1.5), except that AAV2/9-TRE-oChiEF-mCherry or AAV2/9-TRE-NpHR-EGFP was used instead of AAV2/9-TRE-HTG.

For $Ca^{2+}$ recording, same AAVs were injected into vCA1, except that AAV2/9-TRE-GCaMP6s was used instead of AAV2/9-TRE-HTG.

*Retrograde tracing from vCA1 to dCA3*. A Mixture (300 nL) of AAV2/9-CaMKIIα-tTAC and AAV2/9-TRE-oChiEF-mCherry (ratio 1:1) were injected into dCA3 (AP -1.5, LAT ± 2.1, DV -1.7), while RV-BFP-tTAN (500 nL) was injected into vCA1 (AP -3.2, LAT ± 3.4, DV -1.5) simultaneously. In control group, tTAC was omitted for injection.

*Retrograde tracing from downstreams of vCA1*. To define hippocampal cells by two projection targets and CaMKIIα promotor, AAVretro-CaMKIIα-tTAC (200 nL), AAVretro-CaMKIIα-BFP-tTAN (200 nL) were injected into two output regions or co-injected at the following coordinates: mPFC (AP + 1.98, LAT ± 0.6, DV -1.62, angle 10°), NAc (AP + 1.0, LAT ± 1.0, DV -4.2), and amygdala (AP -1.2, LAT ± 3.0,

DV -4.35). AAV2/9-TRE-GCaMP6s (200 nL) were injected into vCA1 (AP -3.2, LAT ± 3.4, DV -3.5). In control groups, either tTAC or tTAN was omitted for injection. All AAVs were co-injected with blue beads. For optogenetic experiments, same AAVs were injected into the same site in vCA1, except that AAV2/9-TRE-GCaMP6s was replaced by AAV2/9-TRE-NpHR-EGFP or AAV2/9-TRE-oChiEF-mCherry.

To define hippocampal cells by four projection targets and CaMKIIα promotor, AAVretro-EF1α-Flpo (200 nL), AAVretro-EF1α-FRT-BFP-tTAN (200 nL), AAVretro-CaMKIIα-cre (200 nL), and AAVretro-EF1α-DIO-tTAC (200 nL) were injected into four regions at the following coordinates respectively: mPFC (AP + 1.98, LAT ± 0.6, DV -1.62, angle 10°), NAc (AP + 1.0, LAT ± 1.0, DV -4.2), amygdala (AP -1.2, LAT ± 3.0, DV -4.35), and lateral septum (AP + 0.5, LAT ± 0.35, DV -2.75). All AAVs were co-injected with blue beads.

To label different types of hippocampal projectors in slice recording, 300 nL red retrobeads (Lumafluor, R-180) and 300 nL green retrobeads (Lumafluor, G-180) were injected into mPFC and amygdala, respectively.

*Anterograde tracing from dCA3 to vCA1.* AAV2/1-EF1α-BFP-tTAN (300 nL) were injected into dCA3 (AP -1.5, LAT ± 2.1, DV -1.7). Then a mixture (400 nL) of AAV2/9-TRE-HTG and AAV2/9-CaMKIIα-tTAC (ratio 1:1) were injected into vCA1 (AP -3.2, LAT ± 3.4, DV -1.5). For controls, either tTAC or tTAN was omitted for injection.

For photometric experiments, same AAVs were injected into dCA3 and vCA1, except that AAV2/1-EF1α-Flpo was used instead of AAV2/1-EF1α-BFP-tTAN, and a mixture (400 nL) of AAV2/8-EF1α-FRT-BFP-tTAN, AAV2/9-CaMKIIα-tTAC and AAV2/9-TRE-GCaMP6s (ratio 1:1:1) were injected into vCA1.

For electrophysiological experiments combined with optogenetics excitation, same AAVs were injected into dCA3 and vCA1 as photometric experiments, except that AAV2/9-TRE-oChiEF-mCherry was used instead of AAV2/9-TRE-GCaMP6s.

For electrophysiological experiments combined with optogenetics inhibition in SOM + interneuron (Supplementary Fig. 6), AAV2/1-EF1α-BFP-tTAN (300 nL) were injected into dCA3. Then a mixture (400 nL) of AAV2/9-TRE-NpHR-EGFP and AAV2/8-EF1α-DIO-tTAC (ratio 1:1) were injected into vCA1 of SOM-ires-cre mice.

To label PV+ interneuron with dCA3 inputs, AAV2/1-EF1α-Flpo was injected into dCA3 of PV-ires-Cre mice. A mixture (300 nL) of AAV2/8-EF1α-DIO-tTAC, AAV2/8-EF1α-FRT-BFP-tTAN and AAV2/9-TRE-GCaMP6s (ratio 1:1:1) were injected in vCA1.

*Combined anterograde and retrograde tracing.* To label RE cells with inputs from mPFC and outputs to vCA1, the AAV2/1-EF1α-BFP-tTAN (300 nL) and AAVretro-CaMKIIα-tTAC (300 nL) were injected into mPFC (AP + 1.98, LAT ± 0.6, DV -1.62, angle 10°) and vCA1 (AP -3.2, LAT ± 3.4, DV -1.5), respectively. The AAV2/9-TRE-oChiEF-mCherry was injected into RE (AP -0.3, LAT ± 1.3, DV -4.5, angle 18°, 300 nL). For controls, either tTAC or tTAN was omitted for injection.

*Local injections in primate cortex.* One cynomolgus monkey was under general anesthesia (isoflurane 1.5~3%) and sterile conditions after subcutaneous injection of atropine sulfate (0.1 mg /kg), intramuscular injection of tiletamine hydrochloride and zolazepam hydrochloride (0.1 mg/kg) and tracheal intubation. The electrocardiogram (ECG), pulse oximeter (SpO2) and end-tidal carbon dioxide ($CO_2$) were monitored continuously. The core temperature was maintained at around 38 °C by an electric blanket. A craniotomy (1.5 × 2.0 cm$^2$) was opened in the skull over the primary visual cortex. The dura was opened to expose the cortex at 4 sites (~5 mm in between). The virus solutions in glass pipettes were injected (30 nL/min) into the cortex at a depth of ~1.5 mm. A mixture (600 nL) of AAV2/9-TRE-oChiEF-mCherry, AAV2/9-CaMKIIα-tTAC and AAV2/9-CaMKIIα-tTAN-BFP (ratio 1:1:1) were injected into two sites. In the other two sites, either AAV2/9-CaMKIIα-tTAC or AAV2/9-CaMKIIα-tTAN-BFP was omitted for injections. The craniotomy was stabilized by sterile plastic sheets during injection, and then was covered with tissue glue and acrylic cement. Animal was treated with antibiotics and analgesia and returned to its home cage after the surgery. Slice recording was performed 6 weeks after the AAV injection.

**Histology and quantification for fluorescence and co-labeling.** One to four weeks after viral injections, mice were transcardially perfused with phosphate-buffered saline (PBS) followed by 4% (w/v) paraformaldehyde (PFA) in PBS. Brains were post-fixed in PFA overnight at 4 °C. and then cut into 80 μm thick coronal slices with a vibratome (Leica, Germany).

For immunostaining of free-floating sections, sections were incubated in blocking solution (3% bovine serum albumin and 0.5% Triton X-100 in PBS) for 30 min at room temperature and then incubated in blocking solution containing goat anti-GFP (to recognize BFP, 1:1000, Abcam, ab6673), rabbit anti-RFP (to recognize mCherry, 1:1000, MBL, PM005) and guinea pig anti-PV (1:500, SYSY, 195004) antibodies over night at 4 °C. Subsequently, sections were washed with PBS three times (10 min each) and incubated in blocking solution for 2 h at room temperature with fluorescent donkey antigoat alexa fluor 488 (1:500, Invitrogen, A11055), goat antirabbit alexa fluor 647 (1:500, Invitrogen, A21245) and goat antiguinea pig alexa fluor 647 (1:500, Invitrogen, A21450). Finally, immuno-labelled sections were rinsed three times with PBS, mounted with an anti-fade mounting media (Solarbio, S2100) dehydrated and coverslipped.

Images from cell cultures were captured using a Nikon Eclipse 80i fluorescence microscope. Images from brain slices were acquired by an Olympus VS120 fluorescence microscope and an Olympus FV3000 confocal microscope (same acquisition settings for one experiment). For analysis of endogenous fluorescence intensities, hippocampal contours were drawn in Fiji. Mean fluorescence value (arbitrary units) of hippocampal slices were measured after background subtraction and normalized by the area size. Typically, 2–10 slices (FOV) per brain were analyzed. For co-labeling analysis, confocal images were imported into Imaris (v7.4.2, Bitplane) for cell counting.

**Western blot.** The transfected HEK293T cells in 24-well-plate were lysated with 200 μl Protein lysis buffer (50 mM Tris-HCl pH 6.8, 2% sodium dodecyl sulfate, 0.1% Bromophenol blue, 10% glycerol, 1% β-Mercaptoethanol), denatured for 10 min at 95 °C before loading. Using a 12% sodium dodecyl sulfate (SDS) poly-acrylamide gel to separate proteins, samples were transferred to a PVDF membrane (Millipore, #IPVH00010). The blotted membranes were blocked with blocking solution containing 5% non-fat milk powder (Sangon Biotech, A600669) in PBS-T (0.1% Tween-20 in PBS buffer) for 60 min, followed by an overnight exposure at 4 °C in the same buffer to the following primary antibodies: (a) rabbit anti-mCherry polyclonal antibody (1:5000, Millipore #AB356482); (b) rabbit anti-GFP polyclonal antibody (1:2000, Proteintech 50430-2-AP); (c) mouse anti-β-Tubulin monoclonal antibody (1:10,000, ProteinTech # 66240-1-Ig). The membranes were washed and incubated for 2 h at room temperature with the corresponding secondary antibodies: (a) HRP-conjugated goat anti-rabbit IgG (1:10,000, ZSGB-Bio #ZB-5301); (b) HRP-conjugated goat anti-mouse IgG (1:10,000, ZSGB-Bio #ZB-5305). Signals were detected and visualized by exposure to Clarity Western ECL Substrate (BioRad, #1705061) and digitized with Tanon 5500 chemiluminescence imaging analysis system. The uncropped scans of blots are provided as a Source Data file.

**Slice electrophysiology.** Mice were first anaesthetized by isoflurane (5%) and then deeply anaesthetized by high-dose isoflurane (~150 μL in a custom nose mask). Mice were transcardially perfused with ice-cold NMDG-based slicing artificial cerebral spinal fluid (ACSF, 93 mM NMDG, 2.5 mM KCl, 1.2 mM NaH₂PO₄, 30 mM NaHCO₃, 20 mM HEPES, 5 mM Sodium ascorbate, 2 mM thiourea, 3 mM Sodium pyruvate, 25 mM D-glucose,12 mM N-Acetyl-L-cysteine, 10 mM MgCl₂, 0.5 mM CaCl₂, oxygenated with 95% O₂/5% CO₂). After the perfusion, the brain was immediately removed and transferred to ice-cold NMDG-based slicing ACSF. Coronal brain slices (300 μm thick) containing hippocampus were prepared using a vibratome (VT-1200S, Leica) in an ice-cold NMDG-based ACSF. Slices were maintained for 12 min at 32 °C in NMDG-based ACSF and were subsequently transferred into HEPES-based solution containing (in mM: 92 NaCl, 2.5 KCl,1.2 NaH₂PO₄, 30 NaHCO₃, 20 HEPES, 5 Sodium ascorbate, 2 thiourea, 3 Sodium pyruvate, 25 D-glucose, 2 MgCl₂, 2 CaCl₂, bubbled with 95% O₂/5% CO₂) at room temperature and incubated more than 1 h before recording, and then were kept at room temperature (20–22 °C) until start of recordings. For the recording, slices were transferred to a recording chamber and infused with ~30 °C recording ACSF (in mM: 119 NaCl, 5 KCl, 1.25 NaH₂PO₄, 26 NaHCO₃, 10 D-glucose, 1 MgCl₂, 2 CaCl₂, bubbled with 95% O₂/5% CO₂). All chemicals were purchased from Sigma-Aldrich.

Patch pipettes (4–7 MΩ) pulled from borosilicate glass (Sutter instrument, BF150-86-10) were filled with a K-gluconate-based internal solution (in mM: 126 K-gluconate, 2 KCl, 2 MgCl₂,10 HEPES, 0.2 EGTA, 4 MgATP₂, 0.4 Na₃GTP, 10 Na-phosphate creatine, 290 mOsm, adjusted to pH 7.2~7.3 with KOH). Whole-cell recording was performed with a Multiclamp 700B amplifier and a Digidata 1440 A (Molecular Device). Voltage-clamp recording was conducted at holding potential −70 mV. Current-clamp recording was conducted at membrane potential −70 mV. For optogenetic stimuli, the blue light pulse (480 nm) from a LED engine (Sola SE5-LCR-V8, Lumencor) was triggered by digital commands from the Digidata 1440 A and delivered through the objective to illuminate the entire field of view. The yellow light from a DPSS laser (589 nm, Shanghai Century) was delivered through optical fiber positioned above the brain slice under the objective. The power at tip of fiber was ~18 mW with ~1 mm distance to the recording site. For the cells expressing TRE-effectors, we only recorded fluorescent cells expressing effector tags unless otherwise stated. The slice data were acquired by pClamp10 (Molecular Device) and analyzed by Clampfit (v10.7).

To obtain the primate brain slices, the cynomolgus monkey was maintained under general anesthesia. The craniotomy was re-opened, and a fresh brain tissue block was cut from an injection site and immediately transferred to ice-cold NMDG-based ACSF. Primate brain slices were prepared similarly as mouse brain slices and recorded in recording ACSF. Animal was monitored continuously until the slice preparation and recording were done for all injected sites, and then was euthanized with an overdose tiletamine hydrochloride and zolazepam hydrochloride.

**Behaviors with photometry and optogenetics.** Four to six weeks after stereotaxic injections, animals were implanted with 200 μm diameter optic fibers (NA 0.37, Anilab, China) in the ventral CA1 (dorsal part: AP -3.2 mm, LAT 3.4 mm, DV -1.2 mm; ventral part: AP -3.2 mm, LAT 3.4 mm, DV -3.3 mm). Animals were singly housed and were allowed one week for recovery after the surgery. Animals were habituated to the experimenter by handling for at least 5 days beforehand, and the behavioral experiments were conducted during the animal's light cycle. For

open field test, animals were placed in an open arena (40 × 40 × 45 cm) and allowed to explore for 15 min.

For conditioned place preference (CPP), the apparatus consists of two square contexts (25 × 25 × 30 cm, connected via a bridge 7 × 10 × 30 cm) with identical pump port on the opposite site but distinct walls and odors. Animals were water restricted and maintained at 90% body weight two days before experiments. During habituation, mice were habituated to the two connected contexts for 15 min without water reward. During the training sessions (day 2–4), mice were first confined in the rewarding context for 10 min (2 µl drop upon licking, one drop per 5 s), and then confined in nonrewarding context for another 10 min (no water). On day 5, animals were allowed to freely explore the two contexts for 15 min without reward (CPP memory retrieval) and then for another 15 min with water reward available. On day 6, animals went through the same procedure as day 5 except that setup orientation was reversed and the reward was changed to sucrose. The CPP score is calculated as (A−B)/(A + B) where A and B are the time that the animal has stayed in the two contexts respectively. For foot shock sessions, animals received six foot-shocks (2 s, 0.7 mA, 60 s interval) in the conditioning context.

For real-time place preference paradigm, the behavioral apparatus was the same as CPP except that the water reward was not available. On day 1, mice were allowed to freely explore the two connected contexts for 15 min. On day 2, a blue laser light (5 ms pulse at 20 Hz, 462 nm diode BLM462TA, Shanghai Lasers, China) or yellow laser (constant, 589 nm DPSS, YL589T6, Shanghai Lasers, China) was delivered into vCA1 when the animal entered and stayed in the stimulation context and was turned off once animal left the stimulation context. The laser intensity was 19–20 mW at the tip of the optic fiber and was controlled by a master-8 (A.M.P.I., Israel). The preference score for the stimulation context is calculated same as CPP score.

For photometry in behaving animals, the optic fibers were connected with a 465 nm blue LED light source (Cinelyzer, Plexon Inc). The excitation light at the tip of fiber was adjusted to be 14–16 µW (setting 10–12 in the LED controller). Emission signals and behavioral videos (640 × 480 pixels at 30 Hz) were captured simultaneously by a multifiber photometry recording system (Cinelyzer, Plexon Inc). The animal trajectories were analyzed and plotted by the Cinelyzer GUI.

Raw data of emission signals in photometry were acquired and directly extracted for analysis in MATLAB (MathWorks) without any filtering. The area (mean) of the $Ca^{2+}$ signals in a defined window were quantified for the event-related responses. The baseline and response periods were divided into equal numbers of bins. The $Ca^{2+}$ signals in each bin were averaged. For non-significant $Ca^{2+}$ response (Mann Whitney U Test), the amplitude was set as zero. As the maximal water/sucrose reward was one drop per 5 s, the shortest trial for water/sucrose reward is 5 s long. Thus, the 0.5 s period before water/sucrose reward onset was computed as the baseline. Based on the decay of $Ca^{2+}$ signals, the 3 s period after sucrose delivery and the 2 s after water delivery were computed as the response. For the foot shocks, the 2 s before foot shock onset was computed as the baseline and the 10 s period after stimulus onset was computed as the response, except 5 s response period in Fig. 6 which showed faster decay. For polar plot, the averaged response for each type of projection cells was computed in each animal. Then the responses were averaged across animals in each projection type and normalized by the maximal response in all projection types. To analyze photometric signals when the animal is moving to the center of the open field (the locomotion is continuous), the filtered $Ca^{2+}$ signal from the Cinelyzer was used (smoothed with an exponentially weighted moving average[61], tau = 0.2 s). The center area is defined as a rectangle with 40% of width and length of the open arena. For each center entry trial, the 2 s before the center entry was computed as the baseline and 2 s after the entry was computed as the signal changes related to the center entry. The same time setting also applies for each exit trial. Short trials (<4 s) were not included for analysis.

**Statistics and reproducibility**. Mean values are accompanied by the standard error of the mean (S.E.M.). For the box plot, data are presented as median (center line) with 25/75 percentile (box) and min and max of the data points (whiskers); the plus indicates mean. No statistical methods were used to predetermine sample sizes.

Statistical analysis was performed in GraphPad Prism 8. The Shapiro-Wilk normality test was tested for all dataset. If the null hypothesis of normal distribution was rejected, non-parametric statistical tests were used. Mann-Whitney U test, Wilcoxon matched-paired signed rank test, two-sided unpaired or paired t-test, and one-way ANOVA test were used to test for statistical significance. Statistical parameters including the exact value of n, precision measures (mean ± SEM) and statistical significance are reported in the text and in the figure legends (see individual sections). The significance threshold was placed at $a = 0.05$ (n.s., $P > 0.05$; *$P < 0.05$; **$P < 0.01$; ***$P < 0.001$; ****$P < 0.0001$).

**Reporting summary**. Further information on research design is available in the Nature Research Reporting Summary linked to this article.

## Data availability

Source data are provided with this paper and publicly available at the following repository https://github.com/XuChunLab/IBIST. The plasmids created in this study are available on Addgene (#172119–172127). Source data are provided with this paper.

## Code availability

The codes used for analysis in this manuscript are available from the following repository https://github.com/XuChunLab/IBIST.

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

## Acknowledgements

We thank all members of the Xu lab and the Long Lab. We thank Drs. Miao He and Xiaohui Zhang for comments on the early version of our manuscript. We thank non-human primate facility for providing a cynomolgus monkey, imaging facility and animal facility for technical support, B. Roska, Z.-K. Qian, H.-K. Zeng, M.-M. Luo and L.-Q. Luo for sharing plasmids, E.M. Callaway for sharing rabies-related cell lines and plasmids, A.-X.C. and W.Z. for technical help. C.X. is supported by National Key R&D Program of China (2020YFE0205900), the Strategic Priority Research Program of the Chinese Academy of Sciences (XDB32010105, XDBS01010100), Shanghai Municipal Science and Technology Major Project (2018SHZDZX05), Lingang Lab (Grant LG202104-01-08), the National Natural Science Foundation of China (31771180 and 91732106) and the international collaborative project (201978677) of Shanghai Science and Technology Committee.

## Author contributions

H.-S.C., X.-L.Z., R.-R.Y., G.L. and C.X. conceived the project. G.L. and C.X. supervised the work in the study. X.-L.Z. designed and tested DNA constructs of split controller, performed Western blot and analyzed the results. R.-R.Y. generated the rabies vectors, performed rabies-related experiments and analyzed the results. H.-S.C., G.-L.W., R.-R.Y., N.Z. and X.-Y.Z. injected the AAV vectors, performed the histology and confocal imaging and analyzed the results. G.-L.W. performed the electrophysiological recording from acute brain slices and analyzed the results. H.-S.C. and S.Q. performed the photometric recording and analyzed the results. Y.-F.X., L.-J.Z. and Z.-M.S. performed the surgery in the primate. D.-Y.W. and X.-H.X. performed AAV injection and the histology and analyzed the results. H.-S.C. and C.X. wrote the manuscript with inputs from all authors.

## Competing interests

The authors declare no competing interests.
