## [Peer Review File · Nature Communications]

An intein-split transactivator for intersectional neural imaging
and optogenetic manipulationREVIEWER COMMENTS

Reviewer #1 (Remarks to the Author):

In the manuscript titled “An intein-split transactivator for intersectional neural imaging and optogenetic manipulation”, the authors developed an intersectional approach to target specific cell populations defined by the expression of intrinsic genetic markers and anatomical connectivity. After validating the system, they used it to label hippocampal cell groups projecting to different target(s) and investigated their calcium responses to rewarding and aversive stimuli. While this manuscript showcases an interesting approach that could be useful for the systems and circuit neuroscience community, it is in its current state overwhelming and overly ambitious with the multitude of IBIST combinations tested but yet lacking in explanation, interpretation, and discussion. This is especially the case in the “application of the system” where key behavioral analyses are missing for their claims and superficial descriptions were provided.

Major concerns:

1. The authors try to accomplish two goals in this study, a) present and validate the IBIST as a tool for circuit neuroscience and b) use the IBIST system to study how different hippocampal neurons behave to relevant stimuli. This not only makes the manuscript overwhelming but unfortunately, none of the 2 goals are comprehensively met. Structuring the study around only one of the two goals would be more focused and convincing: a) present the IBIST and provide extensive validations (see below) or b) use IBIST to thoroughly investigate hippocampal subpopulation in rewarding vs aversive stimuli encoding.

2. Points related to option a)

- The authors claim that IBIST allows the identification and labelling of different subpopulations. Please provide control experiments showing the purity of the “multi-projector cells”. For example, how many of the mPFC/Amy projecting cells actually project to both. In other words, is this subpopulation contaminated by mPFC or Amy projecting cells and by how much?

- What is the quantitative efficiency of IBIST? What proportion of cells within a given subpopulation can it label?

- If IBIST can be used across experimental models, validation experiments should be comprehensive for all model and data analysis should also show that the same efficiency and specificity can be achieved.

- The authors validated different setups of the IBIST system in different brain areas (different parts of the hippocampus in Figure 1G-I and Figure 2A-D). Why these specific regions over others. How applicable is the IBIST in subcortical area, midbrain or basal ganglia structures?

- The authors need to discuss the overall technological feature of IBIST and not only separate

components. Otherwise this fails to show how useful and applicable this new method could be.

- The authors tried to validate the IBIST system to express optogenetic tools in targeted cell populations with patch clamp recordings. It is unclear from the text, figure legends (Figure 2, S5) and methods which cells were patched.

- If the IBIST system is working the way it should, the patched cells with a light induced effect should also express parts of the IBIST system (ie. BFP in the construct with tTAN). Is that the case?

- In the preparation with the expression of partial components of the IBIST system with an IBIST dependent halorhodopsin (a control), patched cells (supposedly not expressing halorhodopsin) observed a light paired reduction in spikes (Figure 2L, S5J). Could the authors provide some explanation and discussion on this result? Furthermore, this finding contradicts what is written in the text (line 195).

- Why is the light-induced spike probability in supposedly ChiEF expressing preparation so low (Figure S5E) as compared to other IBIST preparations?

- Throughout the manuscript, the authors attempted to measure the specificity of the IBIST system by examining the percentage of cells expressing effector reporter protein with or without a component of the IBIST system, however:

- There is a wide range of specificity depending on the setup, from 95% (Figure S2B), 90% (Figure 1I), 67% (Figure S4E), to 78% (Figure S4K). Could the authors provide some explanation and discussion to these discrepancies among setups.
- Given that individual viruses were used to express tTAC and mCherry (Figure S4A-E), cells labelled as mCherry are not necessarily ones infected with tTAC. Therefore, the quantification in S4E should be interpreted with caution and the caveat discussed. It would have been worthwhile to label somatostatin expressing cells directly either with immunohistochemistry (IHC) or in-situ hybridization (ISH).
- The goal in Figure S4H-K was presumably to target Vgat and Vglut2 expressing cells of the preoptic area. It is therefore not sufficient to investigate the specificity of the IBIST with respect to Vgat and not to also Vglut2. A multiplex fluorescent ISH should be conducted in this case rather than a single channel ISH for Vgat.

3. Points related to option b)

- Data in figures 3 to 5 report that the different projectors respond differently to a shock, water or sucrose. However, this is not substantial enough towards the understanding of hippocampal cell populations and their role in salient stimuli encoding:

- What is the relative proportion of each population with respect to the hippocampus? Please quantify the mapping/anatomical IBIST data.
- Besides their distinct projection, how are they functionally or molecularly different? For ex, In vitro electrophysiological characterization of each labelled cell type would be necessary.
- Bidirectional optogenetic manipulation of each projectors would provide information about their role

and contribution to reward- and aversion-related behaviors.

- Please provide control experiments showing the purity of the “multi-projector cells”. For example, how many of the mPFC/Amy projecting cells actually project to both. In other words, is this subpopulation contaminated by mPFC or Amy projecting cells and by how much?

- The authors made a bold statement in the abstract that stated that “hippocampal cells tune differently to distinct emotional valences depending on the pattern of projection targets” (line 36). While the authors showed some interesting results along this line in figures 3-5, there is little interpretation and discussion of these results with respect to the literature. It is hard to gauge the novelty and the significance of these results.

- The authors used the entry to the center of the open field test (Figure 3D) as a measure of anxiety culminating in the interpretation that an increase in calcium activity in the CA3 projecting CA1 cells exhibited anxiety related signals (line 274-275). However, the authors provide no behavioral data to show that the center is anxiogenic/aversive. Please quantify the time spent in the center vs outer region of the arena. Also, the mice should be tested in another anxiety test such as the elevated O-Maze while recording calcium activity. Changes in calcium transients should then be correlated with the location of the mouse in the different arms of the maze. Furthermore, locomotion could also play a role in the change in calcium activity observed. Additional analysis should be done to disprove this point.

- The authors stated throughout the manuscript that the activity of hippocampal neurons projecting to different targets were investigated in a fear conditioning paradigm (Figure 3-5, S3, S6). However, the authors provided no evidence that the mice that underwent this paradigm learned to freeze to the shock paired context and not generalized to other contexts. Please analyse and report the behavioral performance. In the only panels with freezing performance (Figure S6D), it is unclear which animals in which figure(s) they were corresponding to. Yet, there is still no comparison between freezing performances. Unless much more details in the behavioral performance across all groups are provided, it is more appropriate to say that the authors investigated the calcium responses of different hippocampal populations to aversive stimuli such as shocks rather than in a fear conditioning paradigm.

Minor concerns:

1. It seems that in vitro, the rtTA system (Figure S1C) infects much less cells than the tTA (Figure 1D) IBIST system. Could the authors provide some explanation and discussion why this is the case?

2. An anti-GFP antibody is used by the authors to label and enhance the fluorescence of BFP (line 1037, Figure 2). Could the authors provide some literature or control data supporting that this particular antibody is specific to GFP as well as BFP?

3. The tTAN labelling is not visible in Figure 3B (right panel), could the authors provide a better example image?

4. Is the plot in figure 4H quantitative or qualitative? If it is the former, the scale bar for the size of calcium activity between rings is missing.

5. The location of the implants cannot be observed in the example images of Figure 5B, could the authors provide a larger field-of-view where they are visible?
6. Only a superficial description of the fear conditioning paradigm was provided in the methods (line 1151-1152). Could the authors provide a more detailed description? What is the criterion for freezing and which windows were considered?
7. How does the calcium activity of individual animals change in response to cues? It would be worthwhile to show in the heatmaps, the responses of individual cells to each trial rather than only the average response.
8. How does the placement of the implants for fiber photometry influence the signal recorded? It seems that the implants were placed in the same location (Figure 4C) but differently projecting hippocampal cells are spatially segregated. This is especially true for the mPFC and Amy projectors, which seem to be much further away than the other projectors. Are the weak responses to different cues in this population influenced by the distance between the fiber and the location of the majority of the GCaMP labelled cells?
9. Could the authors provide some rationale on the reason for the time periods chosen for the baseline and cue response windows for the fiber photometry experiments (methods, page 46)? Why are these windows different in duration (shock BL 2s, shock response 10s; lick BL 0.5s, lick response 3s)?
10. Why are mice restrained in the non-rewarding chamber of the conditioned place preference task (line 1144)? If the authors are meant to say that mice are physically restricted, please change the wording. Restraint could also mean a stressor applied to the mice. If that is the case, the authors then need to explain why this unconventional addition was made to the CPP.

Reviewer #2 (Remarks to the Author):

Chen et al report the development of a novel system to allow expression of proteins of interests, such as GCaMP, by ingeniously splitting tTA into two halves, allowing intersectional control of protein expression that can be complemented by existing cre-based approaches. Overall, this is a potentially useful system that would be of interest to the readership of the journal provided the revisions below are addressed.

1. The authors mention there is “leak by AAV-tTAN or AAV-tTAC when the titer was higher than 10^{13} ”. Please characterize the leak conditions in more detail. What is the extent of leak in vivo for different amounts of virus concentration? Does this problem vary with cell type of brain region?
2. Please characterize if there is long-term toxicity of tTAN and tTAC across a range of concentrations in vivo and in vitro, by showing if expression of these components alters cell viability, normal expression of other genes and biophysical characteristics.
3. The abstract is not very informative. For example, the word tTA doesn't even appear in the abstract,

even though it is a central part of this system. Furthermore 'intein' is used in the abstract even though the term is only defined later, in the introduction. Please re-write the abstract to make it clearer.

4. Enter addgene identification #s for all constructs deposited

5. Please discuss what are the advantages/disadvantages of this system compared to the existing INTRSECT system from the Deisseroth lab.

6. Every year there are many novel systems neuroscience methods published in high profile journals which promise to solve all sorts of limitations of current methods. However, the vast majority of them do not gain widespread use by the community due to inaccessibility of the necessary expertise and materials. We certainly do not need another such publication. To ensure that this method is widely accessible, can the authors make an arrangement (with addgene or the UNC vector core, for example), to make these vectors commercially available? It is unlikely this system would be used and tested by many labs unless the viruses can be easily purchased. Such accessibility is part of the reason that led to widespread use of optogenetic and DREADDs, for example.

Reviewer #3 (Remarks to the Author):

In this manuscript, the authors developed a novel split tetracycline-controlled transcriptional activation system-based viral strategy; the authors named it IBIST (intein-based intersectional synthesis of transactivator). Briefly, the authors split tTA into N- and C-terminal fragments, thereby generated tTAN and tTAC. Functional tTA can be reconstituted when tTAN and tTAC express in the same cell, through split-intein mediated trans-splicing technique for efficient reconstitution. Therefore, in this system, the expression of the effector depends on the co-existence of tTAN, tTAC, and TRE-effector. The authors constructed a number of viral constructs and thoroughly validated the specificity and efficiency of IBIST in vitro and in vivo. Furthermore, by combining IBIST with DNA recombinase-dependent system, cell type-specific promoters, and anterograde and retrograde viruses, the authors demonstrated that IBIST is able to intersect five features (though 4 of the features are 4 different projection targets). Since an effector under TRE can be easily swapped according to research needs, the IBIST technology could be a valuable and widely applicable tool for the neuroscience field.

This reviewer and the authors noted that, recently, intersectional strategies using multiple DNA recombinases (such as Cre and Flp and Dre) together with Cre/Flp/Dre co-dependent intersectional vectors have made it possible to intersect multiple features for cell-type specific manipulations (Fenno et al, 2020, cited by the authors as reference 11). However, it remains uncertain whether the dual- and triple- dependent vectors can achieve sufficient high level expressions required for manipulating or imaging neurons. Thus, the IBIST tool could still be a useful alternative method.

Below is a list of minor points that should be addressed:

1. Figure 1E, S1E. Fluorescence intensity depends on cells density of imaged areas. The increased fluorescent intensity at 72 hours could be due to cell growth rather than expression level.
2. In Figure 1I, in TRE-GFP vector alone transfection, there was ~10% GFP+ cells. The authors should explain here that the TRE vector can have tTA-independent basal expression.
3. Figure 2E. How many neurons in the primate cortex are infected? Is the infection efficiency very low? Such information is very useful for primate research. If the infection efficiency is very low, such strategy is not useful for functional manipulations of a desired population of neurons.
4. Figure S4. All experiments in this figure are n =1 animal experiment. The authors quantified results using "FOV", field of view. This experiment should be done with n=3 replicates. In this Figure S4 (C-E), the number of GCaMP single positive cells is very high (33.1±2.7%). So much leaky expression! This has to be explained. Also, what percentage of mCherry positive cells are also GCaMP positive? This would inform us the efficiency of this method.
5. Figure 4. dF/F for Water and Sucrose conditions ($dF/F < 1\%$) are significantly smaller than dF/F for Foot Shock ($dF/F 10\%$). Therefore, the authors should not normalize the responses in Figure 4H. A similar quantification of Ca responses shown in Figure 5D need to be shown in Figure 4.
6. Line 340. Projections from a single vCA1 to 4 different brain areas (mPFC, NAc, Amygdala, LS) is interesting. Are there any references that previous demonstrated this multi-target projection?
7. It would be informative for readers to have a discussion about the differences between the INTERSECT method from Deisseroth lab (Fenno et al 2020) and IBIST, perhaps in terms of efficiency and expression levels, ease of vector construction, also the leakiness of TRE vectors.

Response to Reviewers

We thank the reviewers and editors for their constructive comments and helpful suggestions. To address the reviewer comments, we have made lots of efforts to conduct a series of additional experiments and analyses and have substantially revised the manuscript as described below. Reviewer comments are shown in **BLACK**, our responses are shown in **BLUE** and manuscript text changes are shown in **RED**.

Reviewer #1:

In the manuscript titled “An intein-split transactivator for intersectional neural imaging and optogenetic manipulation”, the authors developed an intersectional approach to target specific cell populations defined by the expression of intrinsic genetic markers and anatomical connectivity. After validating the system, they used it to label hippocampal cell groups projecting to different target(s) and investigated their calcium responses to rewarding and aversive stimuli. While this manuscript showcases an interesting approach that could be useful for the systems and circuit neuroscience community, it is in its current state overwhelming and overly ambitious with the multitude of IBIST combinations tested but yet lacking in explanation, interpretation, and discussion. This is especially the case in the “application of the system” where key behavioral analyses are missing for their claims and superficial descriptions were provided.

Major concerns:

1. The authors try to accomplish two goals in this study, a) present and validate the IBIST as a tool for circuit neuroscience and b) use the IBIST system to study how different hippocampal neurons behave to relevant stimuli. This not only makes the manuscript overwhelming but unfortunately, none of the 2 goals are comprehensively met. Structuring the study around only one of the two goals would be more focused and convincing: a) present the IBIST and provide extensive validations (see below) or b) use IBIST to thoroughly investigate hippocampal subpopulation in rewarding vs aversive stimuli encoding.

We thank the reviewer for the comments. We have performed additional experiments and data analysis to address the concerns and questions and updated the discussion accordingly. We believe that the revised manuscript has been significantly improved because of these comments.

2. Points related to option a)

- The authors claim that IBIST allows the identification and labelling of different subpopulations. Please provide control experiments showing the purity of the “multi-projector cells”. For example, how many of the mPFC/Amy projecting cells actually project

to both. In other words, is this subpopulation contaminated by mPFC or Amy projecting cells and by how much?

In response to this question, we have thought about two ways to address the purity of “multi-projector cells”. One way is to label “multi-projector cells” with classical fluorescent dyes and compare with those cells labeled by IBIST tools in the same animal. However, we are not sure about the differences in labeling preference of AAV vectors and dyes which may hinder the estimation of the purity of “multi-projector cells”. The other way is to perform control experiments with omission of retrograde tracing from one of the downstream targets. We undertook the latter way and analyzed control experiments for NAC-amygdala projectors to estimate the specificity of IBIST system. In these control experiments, we consider all the hippocampal cells labeled by the IBIST-dependent effector as potential contaminations for the “multi-projector cells”. We found that virtually no hippocampal cells were labeled in those control experiments (**Figure 4c**, **Figure R1**). These quantifications suggest that the subpopulation contaminated by “single-projector cells” is negligible.

Figure R1. The purity of the IBIST-based labeling of hippocampal projectors. **A**, Box plot of normalized cell number. Kruskal-Wallis test revealed significant differences between groups (tTAN, 0.3 ± 0.1 , $N = 3$ animals; tTAC, 0.1 ± 0.1 , $N = 3$ animals; tTAN+tTAC, 255.4 ± 13.4 , $N = 3$ animals; $P = 4.28 \times 10^{-6}$; Dunn’s multiple comparisons, tTAN+tTAC vs. tTAN, $P = 3.07 \times 10^{-4}$; tTAN+tTAC vs. tTAC, $P = 1.56 \times 10^{-5}$). **B**, Box plot of normalized fluorescent intensity. One-way ANOVA revealed significant differences between groups (tTAN, 3.4 ± 0.3 , $N = 3$ animals; tTAC, 3.7 ± 0.2 , $N = 3$ animals; tTAN+tTAC, 193.0 ± 14.4 , $N = 3$ animals; $F_{(2,30)} = 149.7$, $P = 2.45 \times 10^{-16}$; Turkey’s multiple comparisons, tTAN+tTAC vs. tTAN, $P = 5.16 \times 10^{-14}$; tTAN+tTAC vs. tTAC, $P = 5.76 \times 10^{-14}$).

- What is the quantitative efficiency of IBIST? What proportion of cells within a given subpopulation can it label?

To address this question on IBIST efficiency, we quantified the numbers of tTAN+ cells and GFP+ cells in our validation experiments in **Figure 1**. We calculated the ratio between tTAN+/GFP+ cells and tTAN+ cells as a proxy for the IBIST efficiency, although it could be an underestimation if not all the tTAN+ cells are also expressing tTAC. We found that, among all the tTAN+ cells, majority of cells are positive for the GFP reporter expression

(100% for *in vitro*, $90.9 \pm 3.2\%$ for *in vivo*, **Figure R2**). These results suggest that IBIST tools exhibit high efficiency in driving effector expression.

Figure R2. The efficiency of IBIST system. **A**, Percentage summary of all the tTAN⁺ cells labeled by HTG *in vitro*. (100±0% co-labeled vs. 0±0% tTAN only; Mann-Whitney U test, $P = 0.029$, $n = 4$ FOV). **B**, Percentage summary of all the tTAN⁺ cells labeled by HTG *in vivo*. ($90.9 \pm 3.2\%$ co-labeled vs. $9.1 \pm 3.2\%$ tTAN only; Paired t -test, $P = 5.05 \times 10^{-5}$, $n = 6$ FOV, $N = 3$ animals).

For the subpopulation of hippocampal projection cells, the proportion of IBIST-labeled cells within a given subpopulation is also dependent on the retrograde efficiency of AAV vectors. To estimate that efficiency, we performed additional experiments using a retrograde fluorescent dye as a reference for the viral tracing. We compared the number of cells labeled by the retrobeads and AAVretro vectors, respectively (**Figure R3**). By focusing on the mPFC-projecting hippocampal neurons, we found that the ratio between IBIST vector-label cells and retrobeads-labeled cells is $59.3 \pm 15.5\%$ (**Figure R3**). These results suggest that the efficiency of retrograde AAV vectors is on average about 60% of the efficiency of fluorescent dyes.

Figure R3. Efficiency of IBIST. **A**, Scheme illustrating injections to label mPFC-projecting cells in vCA1v by both AAV and red retrobeads. **B**, Examples showing fluorescence of tTAN (shown in magenta) and retrobeads (shown in green). Left, injection sites. Right, co-localization of tTAN and retrobeads. Scale bars: 50 and 500 μm . **C**, Summary of the tracing ratio. $n = 13$ FOVs, $N = 3$ animals.

- If IBIST can be used across experimental models, validation experiments should be comprehensive for all model and data analysis should also show that the same efficiency and specificity can be achieved.

This is a great suggestion. We have followed this advice and used the fluorescence intensity measurements (e.g. **Figure 1e**) to represent the specificity of IBIST system across all experimental models. We have added the same measurements in other figures including **Figure 1-3** and **Supplementary Figure 4-5**.

- The authors validated different setups of the IBIST system in different brain areas (different parts of the hippocampus in Figure 1G-I and Figure 2A-D). Why these specific regions over others. How applicable is the IBIST in subcortical area, midbrain or basal ganglia structures?

We have chosen different parts of hippocampus because we know the circuit connectivity in these areas in a great detail and it is straightforward for us to access the performance of the IBIST system.

To follow the reviewer's suggestion and extend the IBIST in other brain structures, we have validated the IBIST system in a thalamic nucleus, the nucleus of reunien (RE). This nucleus is a key bridge linking prefrontal cortex with hippocampus (Cassel et al., 2013). Using IBIST tools, we have combined anterograde and retrograde tracing and successfully labeled those RE cells that connect with prefrontal cortex and ventral hippocampus. These results are now shown in **Supplementary Figure 5j-k** and reported in the text (**results: page 10, line 238-246; methods: page 41, line 1036-1041**). Overall, we are convinced that the IBIST system is applicable in many brain structures.

- The authors need to discuss the overall technological feature of IBIST and not only separate components. Otherwise this fails to show how useful and applicable this new method could be.

Thanks for the suggestion. The overall technological feature of the IBIST system is summarized as following: (1) ease to design the IBIST construct, (2) reconstitution of tTA at the protein level, (3) multiplexed number of intersectional features, (4) reversible control by doxycycline, (5) compatibility with a large library of common TRE-effectors, (6) compatibility with Cre and Flp transgenic animals, rabies-based transsynaptic tracing and the INTRSECT system (Fenno et al., 2020) for intersectional control. We have now updated discussions in the text on the technological feature of the IBIST system (**page 15, line 347-353**).

- The authors tried to validate the IBIST system to express optogenetic tools in targeted cell populations with patch clamp recordings. It is unclear from the text, figure legends (Figure 2, S5) and methods which cells were patched.

We apologize for this ambiguity. In the experimental group, we chose fluorescent cells for recording (mCherry for oChiEF, EGFP for NpHR). The only exception is in **Figure 2a-e**. This was the very first optogenetic experiment we did with IBIST tools. We patched cells without looking at their fluorescent markers because most cells at the injection site were well infected by IBIST vectors. In the control group, we chose the cells from the same brain area without checking fluorescent markers, as those cells are not necessary expressing any fluorescent markers. We have updated this information in the text accordingly (**legend: page 20, line 480; methods: page 47, line 1196-1197**).

- If the IBIST system is working the way it should, the patched cells with a light induced effect should also express parts of the IBIST system (ie. BFP in the construct with tTAN). Is that the case?

Yes, the reviewer is correct. To record the light-induced response in cells in the experimental group, we typically patched cells with fluorescence tag for opsins. During the recording, we did not check the BFP signals due to the lack of BFP optical filter in the microscope on the patch-clamp setup. Nevertheless, as the BFP and opsin tag are highly co-localized ($96.2 \pm 1.6\%$, **Supplementary Figure 2**), we do believe that the patched cells with light responses also express parts of the IBIST system. We also performed additional recording in the tTAN+tTAC group to confirm that the light-induced spikes are specific for cells expressing fluorescent tags of oChiEF (**Figure R4**).

Figure R4. Light specifically induced spikes in cells labeled by IBIST tools. **A**, Examples showing mCherry fluorescence in patched cells. Scale bars: 10 μm. **B**, Examples showing whole-cell current-clamp recording from mCherry positive and negative cells in brain slices with blue LED light stimulation (blue line). Animals were injected with AAV-TRE-oChiEF and co-injected with AAVs (CaMKIIα promoter). **C**, Summary of spike probabilities (prob.) in hippocampal cells evoked by whole-field blue LED light from animals injected with AAV-TRE-oChiEF and other AAVs (CaMKIIα promoter): N = 4 animals.

- In the preparation with the expression of partial components of the IBIST system with an IBIST dependent halorhodopsin (a control), patched cells (supposedly not expressing halorhodopsin) observed a light paired reduction in spikes (Figurer 2L, S5J). Could the

authors provide some explanation and discussion on this result? Furthermore, this finding contradicts what is written in the text (line 195).

Thanks for pointing it out. The light-induced reduction in spikes in the tTAC only group is possibly due to the leaky expression of the TRE-NpHR in a few cells. Although there is a stark contrast in the intensity of endogenous fluorescent tag between tTAC+tTAN group and tTAC group (**Figure 2m**), there might be a few cells with leaky expression resulting in light-induced effects in the tTAC group.

To determine how robust and to what extent the light-induced effect in the tTAC group is, we performed additional patch-clamp recording in new brain slices with the experimental settings same as Figure 2k and Supplementary Figure 6g. In these additional recordings, we did not find any light-induced reduction in spikes. These new results are now pooled together with the previous results and updated in the Figures and text. In **Supplementary Figure 6j**, there is no significant light-induced reduction in spikes ($n = 19$ cells, 9.2 ± 1.7 Hz vs. 8.3 ± 1.8 Hz with light). In **Figure 2o**, there is still a significant but very small light-induced reduction in spikes ($n = 11$ cells, 7.7 ± 1.0 Hz vs. 6.5 ± 1.2 Hz with light). Compared to the tTAC+tTAN group, we think that this small reduction is unlikely to cause functional impact. We have updated the **Figure 2** and the text for results (**page 8, line 186-188**) and have discussed the potential leaky expression of TRE effectors in the text (**page 16, line 378-389**).

- Why is the light-induced spike probability in supposedly ChiEF expressing preparation so low (Figure S5E) as compared to other IBIST preparations?

In response to this question, we have performed additional recordings from another 10 cells with the same experimental settings. We still observed a similar level of light-induced spike probability and have updated the **Supplementary Figure 6e** with the pooled data. We think that this relatively low level of light-induced spike probability is likely due to the AAV vectors used in this tracing strategy. It is possible that the Flp-dependent expression and the efficiency of AAV1 anterograde tracing are relatively weak. The combination of both may result in significantly lower expression of tTAN than that driven by local AAV infections (see **Figure 2e**). Nevertheless, there are still more than half of cells showing spikes induced by a relatively strong light. These cells with light-induced spikes, together with more cells with subthreshold depolarization, may be sufficient for some optogenetic manipulations. In the future, we may think of using ultrasensitive ChR2 variant such as ChRger (Bedbrook et al., 2019) in this type of experiments.

- Throughout the manuscript, the authors attempted to measure the specificity of the IBIST system by examining the percentage of cells expressing effector reporter protein with or without a component of the IBIST system, however:

The three points below are addressed together.

- There is a wide range of specificity depending on the setup, from 95% (Figure S2B), 90% (Figure 1I), 67% (Figure S4E), to 78% (Figure S4K). Could the authors provide some explanation and discussion to these discrepancies among setups.
- Given that individual viruses were used to express tTAC and mCherry (Figure S4A-E), cells labelled as mCherry are not necessarily ones infected with tTAC. Therefore, the quantification in S4E should be interpreted with caution and the caveat discussed. It would have been worthwhile to label somatostatin expressing cells directly either with immunohistochemistry (IHC) or in-situ hybridization (ISH).
- The goal in Figure S4H-K was presumably to target Vgat and Vglut2 expressing cells of the preoptic area. It is therefore not sufficient to investigate the specificity of the IBIST with respect to Vgat and not to also Vglut2. A multiplex fluorescent ISH should be conducted in this case rather than a single channel ISH for Vgat.

Thanks for the comments and suggestions. We agree with the reviewer that the ways to access the IBIST specificity in previous Figure S4A-E and H-K were suboptimal. For previous Figures S4C-E, the potential mismatch expression between mCherry and tTAC may underestimate the specificity. For the previous Figure S4H-K, one possibility of the relatively low specificity is that the in situ hybridization for Vgat is incomplete. We agree with the reviewer that in situ hybridization for Vglut2 is also needed to investigate the IBIST specificity in this case. Unfortunately, our in situ hybridization experiment for Vglut2 did not work out due to the technical failures. This prevented us from investigating IBIST specificity with respect to both Vgat and Vglut2. Therefore, we have now removed these results in previous Figure S4A-E and H-K in the revised manuscript.

The rationale for this experiment was to access the IBIST specificity in Cre mice. To keep pursuing this goal, we have performed new experiments using PV-Cre mice in which we have obtained good immunostaining of PV. We found that $81.6 \pm 4.8\%$ IBIST-labeled cells are PV positive (**Supplementary Figure 5d-f**), reflecting a good specificity of IBIST system. These results are now updated in the text (**page 10, line 230-234**).

Furthermore, we accessed the efficiency of IBIST system in this experiment by computing how many tTN+/PV+ cells are expressing IBIST-dependent GCaMP6s (Supplementary Figure 5d-e). We found that $68.2 \pm 3.0\%$ tTN+/PV+ cells were successfully expressing IBIST effectors (**Figure R5**). This may be an underestimated efficiency because it is not necessary all PV+ cells are expressing tTAC.

Taken together, our results suggest that the specificity and efficiency of IBIST system is sufficient for a wide range of practical applications.

A

Figure R5. The IBIST efficiency in PV-Cre mice. A, Percentage summary of GCaMP6s positive cells in the tTAN⁺ /PV⁺ cells. (68.2±3.0% co-labeled vs. 31.8±3.0% PV only; Paired t-test, P = 6.97×10⁻⁵, n = 12 FOV, N = 3 animals).

3. Points related to option b)

- Data in figures 3 to 5 report that the different projectors respond differently to a shock, water or sucrose. However, this is not substantial enough towards the understanding of hippocampal cell populations and their role in salient stimuli encoding:

- What is the relative proportion of each population with respect to the hippocampus? Please quantify the mapping/anatomical IBIST data.

To address this question, we first performed NeuN staining in hippocampal slices in order to estimate the total number of hippocampal neurons. We then computed the relative proportion of each population with respect to the hippocampus as suggested by the reviewer. We found that, among the vCA1 neurons, the amygdala-projecting neurons are the most abundant and the Amy/mPFC-projecting neurons are the least (NAc, 13.1%; Amy, 14.1%; mPFC, 3.0%; NAc/Amy, 8.2%; NAc/mPFC, 5.5%; mPFC/Amy, 2.2%, **Figure R6**).

Figure R6. Proportion of ventral CA1 cells labeled by single or double projections. A, Proportion of cell number defined by single or double retrograde tracing by IBIST.

- Besides their distinct projection, how are they functionally or molecularly different? For ex, In vitro electrophysiological characterization of each labelled cell type would be necessary.

The reviewer has raised an important and interesting question about different types of projecting cells in ventral hippocampus. For the electrophysiological characteristics, we have performed new patch-clamp recording in brain slices from mice combined with retrograde tracing (n = 36 cells, N = 9 mice). In this experiment, we visualized the single- and double-projecting neurons in vCA1 by injecting red and green retrobeads into mPFC and amygdala, respectively. The input-output measurements showed that amygdala-projecting cells were less excitable than mPFC-projecting cells and double-projecting cells (**Figure R7**). The detailed comparison of electrophysiological parameters is summarized in the following table (**Table R1**). While most parameters are indifferent, a few are significantly different. For instance, the input resistance (R_m) is significantly lower in amygdala projectors than in mPFC projectors. These results highlight the importance of a comprehensive study for the molecular and electrophysiological characterization of subtypes of hippocampal projection cells. We will collect more data for other types of projection cells in vCA1 and hopefully report these findings together in a new manuscript.

For the molecular differences, this is a super interesting question and has begun to be addressed by MAPseq recently, a high-throughput sequencing of barcoded neurons (Gergues et al., 2020). This is beyond the scope of current study and worth of an independent project for the molecular characterizations. We hope to utilize the molecular information and design functional experiments with IBIST tools to investigate the correlation between the molecular profile and the physiological functions in future studies.

Figure R7. Electrophysiological characterization of hippocampal cells. **A**, Scheme illustrating injections to label mPFC-projecting cells or Amygdala-projecting cells in vCA1v by green retrobeads or red retrobeads,

respectively. **B**, Examples showing whole-cell current-clamp recording from hippocampal cells in brain slices upon current injections (left, 80 pA; right, 200 pA). **C**, Examples showing fluorescence of red retrobeads and green retrobeads in patched hippocampal cells. Scale bars: 10 μm . **D**, Summary of spike number evoked by current injection in single-region-projecting cells and double-region-projecting cells. (amygdala projectors: $n = 14$ cells; mPFC projectors: $n = 9$ cells; amygdala-mPFC projectors, $n = 11$ cells; $N = 9$ animals).

Table R1 | Electrophysiological parameters of retrobeads-label cells in the ventral CA1

Parameters	Mean \pm S.E.M.			P value		
	mPFC	Amygdala	mPFC and Amygdala	mPFC vs. amygdala	mPFC vs. mPFC and amygdala	Amygdala vs. mPFC and amygdala
Vm (mV)	-65.3 \pm 1.4	-63.9 \pm 0.9	-65.4 \pm 1.0	0.35	0.96	0.26
Rm (m Ω)	153.1 \pm 31.6	95.0 \pm 5.4	104.6 \pm 11.8	0.036	0.14	0.43
Cm (pF)	142.8 \pm 12.4	189.6 \pm 20.0	176.6 \pm 17.5	0.098	0.15	0.64
Tau (ms)	20.3 \pm 3.2	17.6 \pm 1.6	18.4 \pm 2.6	0.42	0.65	0.79
Vth (mV)	-36.3 \pm 1.5	-36.6 \pm 0.8	-34.3 \pm 1.2	0.83	0.31	0.11
Rheobase (pA)	87.8 \pm 16.9	115.0 \pm 8.1	104.6 \pm 17.4	0.12	0.5	0.56
Spike peak (mV)	41.4 \pm 1.9	47.6 \pm 1.0	43.8 \pm 1.1	0.0045	0.27	0.017
Spike amplitude (mV)	77.7 \pm 2.4	84.3 \pm 1.3	78.1 \pm 1.9	0.017	0.91	0.01
Spike half-height duration (ms)	1.4 \pm 0.1	1.3 \pm 0.04	1.4 \pm 0.1	0.3	0.94	0.098
Spike rise (ms)	0.4 \pm 0.04	0.4 \pm 0.02	0.7 \pm 0.3	0.21	0.41	0.24
Spike decay (ms)	2.9 \pm 0.8	1.8 \pm 0.3	1.7 \pm 0.2	0.2	0.14	0.79

- Bidirectional optogenetic manipulation of each projectors would provide information about their role and contribution to reward- and aversion-related behaviors.

Thanks for this great suggestion. Our photometry experiments revealed that the NAc/mPFC-projecting cells preferentially responded to the foot shock and the water omission whereas the NAc/amygdala-projecting cells responded to both the foot shock and the sucrose reward (**Figure 4h**). These results suggest that NAc/mPFC-projecting cells, but not NAc/amygdala-projecting cells, preferentially contribute to aversion-related behaviors. To test their causal roles in the behavior, we have done bidirectional optogenetic manipulations for these two different types of double-projecting cells in a real-time place preference paradigm using the IBIST tools. We found that excitation of NAc/mPFC-projecting cells induced a place aversion and inhibition did the opposite. In contrast, manipulations of NAc/amygdala-projecting cells did not induce any place preferences (**Supplementary Figure 8**). These results are consistent with the notion that vCA1 neurons process the emotional valence depending on their projection patterns. We have now included these results in the text and discussed their implications in projection-pattern-dependent emotion processing (**results: page 13-14, line 312-327; discussion: page 16-17, line 389-401; legend: page 31, line 724-740; methods: page 39, line 1004-1006, page 49, line 1230-1239**).

- Please provide control experiments showing the purity of the “multi-projector cells”. For example, how many of the mPFC/Amy projecting cells actually project to both. In other words, is this subpopulation contaminated by mPFC or Amy projecting cells and by how much?

A similar question was asked previously. Please see the detailed answer to the first point in option (a) by the reviewer.

In brief, we have performed control experiments with omission of retrograde tracing from one of the downstream targets and have quantified the number of cells labeled by the IBIST-dependent effector, which is considered as potential contaminations for the “multi-projector cells”. As shown in the **Figure 4c**, those control experiments failed to label any hippocampal cells. This is in stark contrast with the experimental groups (**Figure R1**). Therefore, we conclude that the subpopulation contaminated by “single-projector cells” is negligible.

- The authors made a bold statement in the abstract that stated that “hippocampal cells tune differently to distinct emotional valences depending on the pattern of projection targets” (line 36). While the authors showed some interesting results along this line in figures 3-5, there is little interpretation and discussion of these results with respect to the literature. It is hard to gauge the novelty and the significance of these results.

Thanks for the comments. The ventral hippocampus is known to connect with a wide range of brain areas such as mPFC, NAc, amygdala, hypothalamus, thalamus (Cenquizca and Swanson, 2007; Pitkanen et al., 2000). There are studies showing that different subpopulations of hippocampal cells project to distinct downstream targets and exert different functions (e.g. BA-projecting neurons for contextual fear, hypothalamus-projecting neurons for anxiety, mPFC-projecting neurons for anxiety, NAc-projecting neurons for goal-directed behavior and cocaine conditioned place preference) (Ciocchi et al., 2015; Jimenez et al., 2018; Xu et al., 2016; Zhou et al., 2019). While most of prior studies focused on projection neurons defined by single target, these projection neurons could well be a mixture of neurons projecting to single and multiple targets. Recent studies reported that vCA1 neurons with multiple projection targets appear to be mediating different physiological functions from those with single targets (Ciocchi et al., 2015). For example, triple-projecting neurons, targeting the mPFC, amygdala, and NAc, are selectively activated for sharp wave/ripples, but not the anxiety or the goal-directed behavior (Ciocchi et al., 2015). However, this type of experiment is technically challenging and it remains largely unknown how hippocampal neurons with other projection patterns process emotional stimuli. Now, the IBIST system we developed would enable us to address these questions more precisely and easily.

In our study, the double-target-projecting vCA1 cells exhibited distinct Ca^{2+} responses to emotional stimuli in a target-dependent manner. In particular, mPFC-NAc-projectors only

responded to foot shocks but not water rewards, whereas NAc-Amy-projectors responded to both (**Figure 4h and i**). These results are consistent with the notion that vCA1 neurons process the emotional valence depending on their projection patterns. We have added more discussion on this in the text (**page 16-17, line 389-401**).

- The authors used the entry to the center of the open field test (Figure 3D) as a measure of anxiety culminating in the interpretation that an increase in calcium activity in the CA3 projecting CA1 cells exhibited anxiety related signals (line 274-275). However, the authors provide no behavioral data to show that the center is anxiogenic/aversive. Please quantify the time spent in the center vs outer region of the arena. Also, the mice should be tested in another anxiety test such as the elevated O-Maze while recording calcium activity. Changes in calcium transients should then be correlated with the location of the mouse in the different arms of the maze. Furthermore, locomotion could also play a role in the change in calcium activity observed. Additional analysis should be done to disprove this point.

In response to this concern, we first quantified the time that animal spent in the center and outer region. The time in the center was significantly less than that in the outer ($8.2 \pm 1.0\%$ vs $91.8 \pm 1.0\%$, **Figure 3e**), indicating the center is anxiogenic.

To further investigate the specificity of correlation between the increase in Ca^{2+} signals and entering the center, we compared those with the Ca^{2+} signals when the animal was leaving the center. Interestingly, we found that Ca^{2+} signals did not increase when the animal was leaving the center (**Figure 3f and g**).

To investigate the role of locomotion in the change of Ca^{2+} activity as the reviewer suggested, we compared the running speed during center entry with that during center exit. The speed dynamics in both situations turned out to be very similar (**Figure 3f**), indicating that locomotion is not the cause for the specific increase in Ca^{2+} signals upon center entry.

In sum, these additional analyses suggest that the Ca^{2+} signals in CA3-targeted CA1 cells are very likely to reflect the anxiogenic state. We have now included these results in Figure 3 and updated the text accordingly (**page 11, line 260-263**).

- The authors stated throughout the manuscript that the activity of hippocampal neurons projecting to different targets were investigated in a fear conditioning paradigm (Figure 3-5, S3, S6). However, the authors provided no evidence that the mice that underwent this paradigm learned to freeze to the shock paired context and not generalized to other contexts. Please analyse and report the behavioral performance. In the only panels with freezing performance (Figure S6D), it is unclear which animals in which figure(s) they were corresponding to. Yet, there is still no comparison between freezing performances. Unless much more details in the behavioral performance across all groups are provided, it is more appropriate to say that the authors investigated the calcium responses of different

hippocampal populations to aversive stimuli such as shocks rather than in a fear conditioning paradigm.

It is true that we did not measure the fear retrieval, although the animals did go through a fear conditioning protocol. This is because we primarily focused on the Ca²⁺ response of hippocampal cells to various emotional stimuli in this study. In response to the reviewer comments, we opted not to say “a fear conditioning paradigm” in the manuscript. We have now updated the figures (**Figure 4b and Supplementary Figure 7**) and text accordingly (**page 13, line 298**).

Minor concerns:

1. It seems that *in vitro*, the rtTA system (Figure S1C) infects much less cells than the tTA (Figure 1D) IBIST system. Could the authors provide some explanation and discussion why this is the case?

Thanks for pointing it out. The experiments in Figure 1d and Supplementary Figure 1c were done at different times with some different conditions. In particular, the quantity of effector plasmid in Figure 1d was higher than that in Supplementary Figure 1c (1 µg TRE-HTG vs. 0.5 µg TRE-mCherry). Therefore, the fewer cells labeled *in vitro* by the rtTA system was likely due to the quantity of plasmids used in the transfection with the same post transfection time.

2. An anti-GFP antibody is used by the authors to label and enhance the fluorescence of BFP (line 1037, Figure 2). Could the authors provide some literature or control data supporting that this particular antibody is specific to GFP as well as BFP?

The BFP DNA just differs from GFP DNA by a few point mutations (Olenych et al., 2007; Patterson et al., 1997). Thus, both BFP and GFP have common epitopes that GFP antibody would easily recognize. The GFP antibody's cross-reaction with BFP is explicitly stated on a product webpage at Abcam (<https://www.abcam.com/GFP-antibody-ab13970.html>). We used a GFP antibody raised in goat from the same company (ab6673), which has the same immunostaining specificity. Therefore, we have successfully amplified the BFP signals using the GFP antibody (**Supplementary Figure 2**).

3. The tTAN labelling is not visible in Figure 3B (right panel), could the authors provide a better example image?

Done, thank you.

4. Is the plot in figure 4H quantitative or qualitative? If it is the former, the scale bar for the size of calcium activity between rings is missing.

In the original Figure 4H, the Ca²⁺ responses to the same type of stimuli was normalized by the maximal magnitude across animals. Thus, the plot is quantitative with same scale for all type of stimuli after normalization.

To make a better presentation of the Ca²⁺ imaging data (**also suggested by reviewer #3**), we have now separately presented the summary of Ca²⁺ response to each type of stimuli without normalization. We have also performed additional experiments to increase the number of animals for each type of hippocampal projection cells (N = 6 animals in each group, except N = 5 animals in NAc and mPFC-Amy groups, **Figure 4h**). The legend in the text is updated accordingly (**page 24-25, line 562-582**).

5. The location of the implants cannot be observed in the example images of Figure 5B, could the authors provide a larger field-of-view where they are visible?

Done, thank you.

6. Only a superficial description of the fear conditioning paradigm was provided in the methods (line 1151-1152). Could the authors provide a more detailed description? What is the criterion for freezing and which windows were considered?

To follow the reviewer's suggestion in a previous point, we have removed the text describing fear conditioning paradigm. Thus, the detailed description for the fear conditioning paradigm and freezing analysis requested here is not needed in the revised manuscript.

7. How does the calcium activity of individual animals change in response to cues? It would be worthwhile to show in the heatmaps, the responses of individual cells to each trial rather than only the average response.

In response to the question, we have now plotted the Ca²⁺ activity of individual animals in Figure 3-5. The Ca²⁺ signals of single trials were already shown in heatmap examples (**Figure 3i, Figure 4d-g, Figure 5c**). The responses of individual cells were not available in our photometric recording, which represents a population activity instead.

8. How does the placement of the implants for fiber photometry influence the signal recorded? It seems that the implants were placed in the same location (Figure 4C) but differently projecting hippocampal cells are spatially segregated. This is especially true for the mPFC and Amy projectors, which seem to be much further away than the other projectors. Are the weak responses to different cues in this population influenced by the distance between the fiber and the location of the majority of the GCaMP labelled cells?

To address the question by the reviewer, we first reconstructed all the fiber tips in the new **Supplementary Figure 3** based on the histology results and confirmed that the fiber tips for photometry recording were consistently localized at a defined region and there were a number of GCaMP6s-expressing cells nearby (**Figure 4c**).

Next, we plotted the amplitude of Ca²⁺ responses in mPFC-Amy projectors as a function of the distance between the fiber tip and GCaMP+ cell bodies. We failed to see any

obvious correlation between the exact distance and Ca^{2+} responses (**Figure R8**). These results indicate that the Ca^{2+} responses in our study cannot be simply explained by the distance between the fiber tip and location of GCaMP6+ cells and rather reflect the functional characteristics of those GCaMP+ cells.

Figure R8. The relation between optical fiber location and Ca^{2+} response. **A**, Correlation between the Ca^{2+} response amplitude (left, water and sucrose reward; right, two series of foot shocks) and the distance between fiber tips and GCaMP6+ cell bodies.

9. Could the authors provide some rationale on the reason for the time periods chosen for the baseline and cue response windows for the fiber photometry experiments (methods, page 46)? Why are these windows different in duration (shock BL 2s, shock response 10s; lick BL 0.5s, lick response 3s)?

The foot shocks were able to elicit strong and long-lasting Ca^{2+} response. Thus, the baseline was set as 2 s and the response window is set as 10 s. In the rewarding context, the water/sucrose was available as one drop per 5 s. This was aimed to collect many trials of Ca^{2+} responses to small rewards by avoiding continuous licking behavior. Thus, each trial of reward licking was 5 s long in minimum. In other words, the shortest trial for water/sucrose reward is 5 s long. Thus, the 0.5 s period before water/sucrose reward onset was computed as the baseline and the 3 s period after stimulus onset was computed as the response. We have also added this explanation into the methods description (**page 50, line 1250-1253**).

10. Why are mice restrained in the non-rewarding chamber of the conditioned place preference task (line 1144)? If the authors are meant to say that mice are physically restricted, please change the wording. Restraint could also mean a stressor applied to the mice. If that is the case, the authors then need to explain why this unconventional addition was made to the CPP.

We apologize for the misunderstanding. What we meant is that the animal was only accessible to one context during the conditioning but it could freely explore the context without any restriction. This is similar as classical CPP paradigm. We have replaced the word “restrained” with “confined” in the text for clarity (**page 48, line 1220-1223**).

Reviewer #2:

Chen et al report the development of a novel system to allow expression of proteins of interests, such as GCaMP, by ingeniously splitting tTA into two halves, allowing intersectional control of protein expression that can be complemented by existing cre-based approaches. Overall, this is a potentially useful system that would be of interest to the readership of the journal provided the revisions below are addressed.

We appreciate these encouraging comments and constructive suggestions. In the revised manuscript, we performed new experiments and additional data analysis as suggested.

1. The authors mention there is “leak by AAV-tTAN or AAV-tTAC when the titer was higher than 10^{13} ”. Please characterize the leak conditions in more detail. What is the extent of leak in vivo for different amounts of virus concentration? Does this problem vary with cell type of brain region?

To follow the reviewer’s suggestion, we have systematically measured the leak in pyramidal cells by AAV-tTAN or AAV-tTAC with titers at the level of 10^{12} and 10^{13} , respectively. We found that AAV-tTAN at titer of 10^{13} resulted in a low level of leaky expression of effector – TRE-GCaMP6s (**Figure R9**). This leaky expression is most likely attributed to the high-titer of tTAN or tTAC because AAV-TRE-GCaMP6s itself does not have detectable leaky expression at titer of 10^{12} (**Figure R9**). Overall, AAVs with titers at the level 10^{12} virtually all showed undetectable leaky expression. Therefore, we recommend a general guideline to use the IBIST tools with titer lower than 10^{13} in the discussion (**page 16, line 378-382**).

To address the question concerning cell types, we have utilized IBIST-based AAVs with titer of 10^{12} to successfully label pyramidal cells, PV cells and SOM cells, respectively (**Figure 2i, Supplementary Figure 5e and Supplementary Figure 6h**). Thus, we believe that the leaky problem can be overcome by relatively lower AAV titer and this solution is applicable across different cell types.

Figure R9. The Leaky expression of effectors by high titers of IBIST vectors. A, Summary of normalized cell number under different virus titers of AAVs. Gray bars, titer 10^{12} ; Black bars, titer 10^{13} . Unpaired *t*-test;

Reporter only, 0.4 ± 0.4 , N = 2 animals; tTAN ctrl: titer 10^{12} , 1.5 ± 0.8 , N = 4 animals; titer 10^{13} , 88.0 ± 13.8 , N = 5 animals; titer 10^{12} vs. titer 10^{13} , $P = 8.92 \times 10^{-4}$; tTAC ctrl: titer 10^{12} , 0.8 ± 0.3 , N = 3 animals; titer 10^{13} , 0.4 ± 0.3 , N = 3 animals; titer 10^{12} vs. titer 10^{13} , $P = 0.42$; tTAN and tTAC, 348.3 ± 21.9 , N = 4 animals; tTAN and tTAC vs. tTAN ctrl titer 10^{13} , $P = 1.55 \times 10^{-5}$. **B**, Summary of normalized fluorescent intensity. Unpaired *t*-test; Reporter only, 3.2 ± 0.3 , N = 2 animals; tTAN ctrl: titer 10^{12} , 1.3 ± 0.1 ; titer 10^{13} , 36.4 ± 7.9 ; titer 10^{12} vs. titer 10^{13} , $P = 0.0058$; tTAC ctrl: titer 10^{12} , 4.8 ± 1.9 ; titer 10^{13} , 2.5 ± 0.7 ; titer 10^{12} vs. titer 10^{13} , $P = 0.32$; tTAN and tTAC, 129.4 ± 22.3 ; tTAN and tTAC vs. tTAN ctrl titer 10^{13} , $P = 0.0035$.

2. Please characterize if there is long-term toxicity of tTAN and tTAC across a range of concentrations in vivo and in vitro, by showing if expression of these components alters cell viability, normal expression of other genes and biophysical characteristics.

In response to the concern, we performed new experiments in animals 1-3 months post AAV or saline injections into the hippocampus. We examined the number of NeuN positive cells as a proxy for cell viability and found no difference in the neuron number between the AAV injected animals and the saline injected animals (**Figure R10A-C**).

The input-output relationship represents one of most important aspects in the neuronal excitability and functions. At 1-2 months post injections, there were no difference in this I-V curve between AAV and saline injected animals. At 3 months post injection, the I-V curve was increased by a small degree (**Figure R10D-E**).

We also examined a number of biophysical parameters by patch-clamp recording in brain slices. At 1-2 months post injections, there were almost no difference in the biophysical characteristics between AAV and saline injected animals (**Figure R10, Table R2**). At 3 months post injection, there was a small but significant difference in the input resistance (R_m), which may explain the changes in the input-output relationship (**Figure R10E**). Taken together, these results suggest that there are only minor changes of biophysical characteristics in neurons with the long-term expression of tTAN and tTAC. In particular, no changes of I-V curves were observed at 2 months of post-injection, during which most functional studies can already be finished.

Figure R10. Toxicity of IBIST AAV vectors. **A**, Scheme illustrating bilateral injections of IBIST viruses and saline in vCA1. **B**, Examples showing fluorescence of NeuN (shown in green) and mCherry signals. Scale bars: 100 μ m. **C**, Summary of normalized number of NeuN⁺ cells in IBIST (red) and saline control (black). (1 month, 2055 \pm 290.3 Saline vs. 1948 \pm 252 IBIST; Mann Whitney U-test, $P > 0.99$; 2 months, 3148 \pm 67.7 Saline vs. 3167 \pm 161 IBIST; Unpaired t -test, $P = 0.92$; 3 months, 2189 \pm 206.6 Saline vs. 2527 \pm 192.2 IBIST; unpaired t -test, $P = 0.3$; $n = 3$ FOV in each group, $N = 1$ mouse for each group). **D**, Examples showing whole-cell current-clamp recording from hippocampal cells in brain slices upon current injections. **E**, Summary of spike number evoked by current injection. Mann Whitney U-test; 3 months: 110 pA current, 11.1 \pm 1.7 Saline vs. 17.7 \pm 2.9 IBIST, $P = 0.042$; 120 pA current, 12.3 \pm 1.7 Saline vs. 19.6 \pm 3.1 IBIST, $P = 0.029$; 130 pA current, 13.8 \pm 1.8 Saline vs. 20.7 \pm 3.2 IBIST, $P = 0.042$; 10 cells vs. 12 cells, $N = 3$ mice for 1 month; 13 cells vs. 10 cells, $N = 6$ mice for 2 months; 20 cells vs. 7 cells, $N = 5$ mice for 3 months.

Table 2 | Toxicity of IBIST viruses

Parameters	1 month			2 months			3 months		
	mCherry	Saline	P value	mCherry	Saline	P value	mCherry	Saline	P value
N (animal number)	3	3		6	4		3	5	
V _m (mV)	-58.9±1.5	-63.5±1.6	0.05	-54.8±2.3	-62.3±0.7	0.0023	-55.1±3.3	-63.2±0.7	0.0012
R _m (mΩ)	129.6±12.4	109.8±10.2	0.24	135.8±18.2	126.3±10.9	0.64	155.1±12.6	116.8±7.1	0.012
C _m (pF)	142.1±14.9	136.3±11.0	0.76	137.8±19.3	111.8±7.5	0.18	98.5±11.8	138.1±11.6	0.071
Tau (ms)	16.8±1.2	14.3±1.0	0.13	16.1±1.4	13.4±0.8	0.081	15.0±1.7	14.9±0.7	0.93
V _{th} (mV)	-40.1±1.0	-41.7±1.6	0.4	-40.4±2.1	-43.9±1.1	0.12	-40.7±0.7	-39.7±0.9	0.53
rheobase (pA)	71.7±5.8	82.0±16.6	0.53	79.0±15.7	67.7±7.6	0.49	54.3±5.7	73.5±7.7	0.17
spike peak (mV)	43.7±1.4	46.2±1.1	0.18	44.5±2.5	46.1±1.1	0.54	42.6±2.5	45.5±1.9	0.41
spike amplitude (mV)	83.8±2.0	87.9±1.9	0.16	84.9±2.1	90.0±1.7	0.066	83.2±2.7	85.2±2.3	0.65
spike half-height duration (ms)	1.1±0.03	1.1±0.04	0.68	1.2±0.1	1.2±0.02	0.8	1.2±0.1	1.4±0.1	0.13
spike rise (ms)	0.3±0.01	0.3±0.02	0.65	0.3±0.01	0.3±0.01	0.91	0.3±0.02	0.4±0.03	0.19
spike decay (ms)	1.2±0.1	1.2±0.1	0.98	1.3±0.1	1.4±0.02	0.65	1.2±0.1	1.8±0.2	0.11

3. The abstract is not very informative. For example, the word tTA doesn't even appear in the abstract, even though it is a central part of this system. Furthermore 'intein' is used in the abstract even though the term is only defined later, in the introduction. Please re-write the abstract to make it clearer.

We have followed the suggestions by the reviewer and modify the abstract and the text accordingly.

The 'intein' are intervening protein domains that catalyze their own excision from the host protein during protein splicing. This term is widely known in the molecular biology (Vila-Perello and Muir, 2010), therefore, we directly use its name in the title to save the space and have now introduced the term of 'intein' in the abstract. To make the abstract clearer, we removed the first sentence of original abstract and rewrote the summary on the contrasting Ca²⁺ response to emotional stimuli in different subpopulations of hippocampal projection neurons. We have also modified the text in the introduction to better explain the term of 'intein' (page 4, line 82-84).

The tTA is the abbreviation for transactivator. We did not introduce this abbreviation until in the introduction section because of the space constrains in the title and abstract. Therefore, we have kept the full name 'transactivator' in the title and abstract.

4. Enter addgene identification #s for all constructs deposited

We have deposited plasmids at Addgene (#172119 – 172127) as suggested. Thank you.

5. Please discuss what are the advantages/disadvantages of this system compared to the existing INTRSECT system from the Deisseroth lab.

To follow the reviewer's suggestion, we compared our IBIST system with the INTRSECT system from the Deisseroth lab. The main points are following.

- 1) **Commonality of effector vectors.** The effector vector is one of the essential parts for the intersectional strategy. For a given category of experiment (e.g. optogenetic excitation), the IBIST system would only requires one common effector vector, which can be well validated across brain regions and laboratories. However, the INTRSECT system would require various types of effector vectors depending on the specific intersectional strategies. While the cell types can be typically defined by various promoters, Cre and Flp etc., each specific intersection (e.g. combining 2 – 4 conditions) would require a specific vector. Thus, compared to the IBIST system, the commonality of effector vectors in the INTRSECT system is much lower and it is quite laborious to validate effector vectors widely across laboratories.
- 2) **Ease of construct design.** The intersection in the IBIST system occurs at the protein level while that in the INTRSECT system occurs at the DNA level. Therefore, it is more straightforward and simpler for the IBIST system to design the construct in response to controllers. For instance, in the IBIST system, the DNAs of tTAN and tTAC is short in length. They can not only be designed separately, but also fit better for the size limit in the AAV plasmids.
- 3) **Number of intersectional conditions.** We have demonstrated that the IBIST system can intersect more features than the INTRSECT system (Figure 4 and 5).
- 4) **The library of effector vectors.** The TRE effectors already have a large library for various experimental needs. Thus, the IBIST system already has many effector vectors readily usable or testable. For example, AAV-TRE-HTG from Liqun Luo's lab (Miyamichi et al., 2011) work well in our hands. It is not only useful as a fluorescent marker, but also useful as a helper AAV for rabies-based transsynaptic tracing. In contrast, the INTRSECT system continuously need expertise to design and test various effector vectors.
- 5) **Sufficiency of gene expression.** Inserting multiple non-coding DNA sequences into the open reading frame of the plasmid could make it vulnerable to insufficient gene expression for optogenetic manipulation and Ca²⁺ imaging. On top of that, there are considerable variabilities across different AAV serotypes with the same plasmid. Therefore, the chance that we fail to find usable effectors in the INTRSECT system for certain specific intersections could be high. In contrast, it is more likely to succeed with the IBIST system.
- 6) **Leaky expression of the effector vectors.** The TRE effectors for the IBIST system may have tTA-independent basal expression. For example, we have seen that AAV-TRE-mCherry showed a high level of tTA-independent basal expression whereas AAV-TRE-oChIEF-mCherry did not. If the experimental needs are mostly for fluorescent labeling,

some TRE effectors may not be usable. In contrast, the effectors in the INTRSECT system are typically dependent on the Cre and Flp, which are tighter and should have very low basal expression. From the practical point of view, we could find alternative TRE effectors for those leaky effectors to work with the IBIST system. But the INTRSECT system may offer advantage for pure fluorescent labeling, albeit it remains to be shown. Therefore, we do think the power of the IBIST system lies more in intersectional experiments that need high level of gene expression.

Last but not the least, we think that the IBIST system can be combined with the INTRSECT system for sophisticated experimental design. It is totally possible to use the INTRSECT system to control the expression of tTAC and tTAN and then use the IBIST system to achieve sufficient gene expression for optogenetics and Ca²⁺ imaging. Therefore, our IBIST system brings in many more possibilities in the intersectional strategies, which can be realized in a wide range of experimental setups.

We have now updated relevant discussions in the text (**page 15-17, line 347-353, 362-364**).

6. Every year there are many novel systems neuroscience methods published in high profile journals which promise to solve all sorts of limitations of current methods. However, the vast majority of them do not gain widespread use by the community due to inaccessibility of the necessary expertise and materials. We certainly do not need another such publication. To ensure that this method is widely accessible, can the authors make an arrangement (with addgene or the UNC vector core, for example), to make these vectors commercially available? It is unlikely this system would be used and tested by many labs unless the viruses can be easily purchased. Such accessibility is part of the reason that led to widespread use of optogenetic and DREADDs, for example.

We cannot agree more with the reviewer. We have made our best efforts to share the IBIST plasmids and help worldwide use of the viral vectors. To follow the reviewer's suggestion, we have deposited IBIST plasmids at Addgene for distribution to the research community (https://www.addgene.org/Chun_Xu/, #172119 – 172127, mentioned in the text **page 36, line 933-934**). We have also sent some plasmids to a company (Taitool, Shanghai; www.taitool.com) to produce AAV vectors (international orders available at xudf@taitool.com). The related information will be updated at a public webpage <https://github.com/XuChunLab/IBIST> and has been mentioned in the methods section of our manuscript (**page 38, line 975-976**). We are also granting permission to more AAV companies to produce those vectors for the research community. More information about other AAV vendors will be added in the public webpage. We are also setting up some collaborations with laboratories in US and Europe to test and use these viral vectors.

Reviewer #3:

In this manuscript, the authors developed a novel split tetracycline-controlled transcriptional activation system-based viral strategy; the authors named it IBIST (intein-based intersectional synthesis of transactivator). Briefly, the authors split tTA into N- and C-terminal fragments, thereby generated tTAN and tTAC. Functional tTA can be reconstituted when tTAN and tTAC express in the same cell, through split-intein mediated trans-splicing technique for efficient reconstitution. Therefore, in this system, the expression of the effector depends on the co-existence of tTAN, tTAC, and TRE-effector. The authors constructed a number of viral constructs and thoroughly validated the specificity and efficiency of IBIST in vitro and in vivo. Furthermore, by combining IBIST with DNA recombinase-dependent system, cell type-specific promoters, and anterograde and retrograde viruses, the authors demonstrated that IBIST is able to intersect five features (though 4 of the features are 4 different projection targets). Since an effector under TRE can be easily swapped according to research needs, the IBIST technology could be a valuable and widely applicable tool for the neuroscience field.

This reviewer and the authors noted that, recently, intersectional strategies using multiple DNA recombinases (such as Cre and Flp and Dre) together with Cre/Flp/Dre co-dependent intersectional vectors have made it possible to intersect multiple features for cell-type specific manipulations (Fenno et al, 2020, cited by the authors as reference 11). However, it remains uncertain whether the dual- and triple- dependent vectors can achieve sufficient high level expressions required for manipulating or imaging neurons. Thus, the IBIST tool could still be a useful alternative method.

We appreciate these insightful comments and constructive suggestions. In the revised manuscript, we have performed additional data analysis and provided more discussions as suggested.

Below is a list of minor points that should be addressed:

1. Figure 1E, S1E. Fluorescence intensity depends on cells density of imaged areas. The increased fluorescent intensity at 72 hours could be due to cell growth rather than expression level.

We thank the reviewer for pointing out this possibility. To control this change, we normalized the fluorescent intensity by the cell number in Figure 1e and Supplementary Figure 1e. We found that the fluorescent intensity at 72 h is still significantly higher than that at 48 h (**Figure R11**). These results suggest that the continuous expression of TRE-effectors significantly contributed to the increase of the fluorescence intensity, demonstrating the fast expression and high efficiency of the IBIST system.

Figure R11. Increasing expression of IBIST effectors in vitro. **A and B**, Summary of normalized fluorescent intensity at 48 h and 72 h post transfection (n = 4 FOV) in tTA-dependent manner (**A**, 48 h vs. 72 h, 0.042 ± 0.001 vs. 0.07 ± 0.008 , Mann-Whitney U-test, $P = 0.029$) and rtTA-dependent manner (**B**, 48 h vs. 72 h, 0.021 ± 0.003 vs. 0.11 ± 0.018 , unpaired *t*-test, $P = 0.0032$).

2. In Figure 1l, in TRE-GFP vector alone transfection, there was ~10% GFP+ cells. The authors should explain here that the TRE vector can have tTA-independent basal expression.

Thank you for raising this point. The Figure 1l shows the results from animal injected with complete IBIST vectors and TRE vectors. It is aimed to determine the specificity of the IBIST system by quantifying how much degree the fluorescence of TRE vectors depends on IBIST vectors. As the reviewer pointed out, there was ~10% GFP+ only cells among all the GFP+ cells, implying ~10% tTA-independent basal expression of TRE vectors. However, we did not see such degree of leaky expression of TRE vectors in control animals with the same injection of TRE vectors and incomplete injection of IBIST vectors. Thus, we propose an alternative possibility that tTAN+ expression is underestimated because the BFP in some cells is too weak/vague to detect. Along this line, we have observed much higher overlap ratio when we utilized the immunostaining to amplify the BFP signals (**Supplementary Figure 2**). Nevertheless, one should be cautious about tTA-independent leaky expression of TRE vector itself. This should be tested before use of specific TRE vectors and estimated to how much degree it might be. We have pointed this out in the discussion (**page 16, line 378-387**).

3. Figure 2E. How many neurons in the primate cortex are infected? Is the infection efficiency very low? Such information is very useful for primate research. If the infection efficiency is very low, such strategy is not useful for functional manipulations of a desired population of neurons.

These are interesting and important questions. Our intention was only to infect a small area of cortex and record a few of them to verify the applicability of the IBIST system in primate brain. Nevertheless, we quantified the density of infected neurons and found that the density of fluorescent neurons is $899 \pm 162/\text{mm}^2$ (**Figure R12**). While the

functional manipulations of a desired population of neurons depends on both density and total number of infected neurons, the infection efficiency for IBIST in primate brain is certainly optimizable by adjusting the injection volume and number of injection sites.

Figure R12. IBIST-based labeling in primate cortex. **A**, Example picture showing fluorescence of oChiEF (shown in green) in primate cortex. Scale bar: 100 μ m. **B**, Summary of normalized cell number. 898.6 ± 162.2 cells were labeled by IBIST ($n = 6$ FOV, $N = 1$ monkey).

4. Figure S4. All experiments in this figure are $n = 1$ animal experiment. The authors quantified results using “FOV”, field of view. This experiment should be done with $n=3$ replicates.

To follow the reviewer’s suggestion, we have performed additional experiments to increase the animal number and substantially revised this figure based on the reviewers’ comments (panel a-c, $N = 3$ animals; panel d-f, $N = 3$ animals; panel g-i, $N = 4$ animals, panel j & k, $N = 4$ animals). This updated figure is now referred as **Supplementary Figure 5**.

In this Figure S4 (C-E), the number of GCaMP single positive cells is very high ($33.1 \pm 2.7\%$). So much leaky expression! This has to be explained. Also, what percentage of mCherry positive cells are also GCaMP positive? This would inform us the efficiency of this method.

Thanks for the comments and suggestions. We agree with the reviewer that the IBIST specificity in Figure S4C-E is obviously suboptimal. Reviewer 1# also raised a similar question and suggested that mCherry positive cells are not necessary equal to tTAC-expressing cells in SOM-Cre mice. Thus, it is better to use immunostaining to visualize the SOM positive cells. However, the immunostaining of SOM in our hands is apparently incomplete as indicated by the uneven distribution of signals in the neighboring areas and depths of brain slice. To better access the specificity in Cre mice, we have performed new experiments using PV-Cre mice in which we have obtained good immunostaining of PV. These results are now shown in the **Supplementary Figure 5d-f**. We found that $81.6 \pm 4.8\%$ of IBIST-labeled cells are PV positive, reflecting a good specificity of IBIST system.

To access the efficiency of IBIST system in this experiment, we computed how many tTN+/PV+ cells are expressing IBIST effector as a proxy for the IBIST efficiency. Our results

suggest that $68.2 \pm 3.0\%$ of PV+ cells were successfully expressing IBIST effectors (see **Figure R5** for the response to reviewer 1#), reflecting a ~70% efficiency of IBIST system in the experimental strategy using Cre mice.

5. Figure 4. dF/F for Water and Sucrose conditions (dF/F < 1%) are significantly smaller than dF/F for Foot Shock (dF/F 10%). Therefore, the authors should not normalize the responses in Figure 4H.

A similar quantification of Ca responses shown in Figure 5D need to be shown in Figure 4.

We have followed the suggestion and separately presented the summary of Ca²⁺ response to each type of stimuli in Figure 4 without normalization, in a similar way as in Figure 5d. We have also performed additional experiments to increase the number of animals for each type of hippocampal projection cells in **Figure 4h**. The legend is updated accordingly (**page 24-25, line 562-582**).

6. Line 340. Projections from a single vCA1 to 4 different brain areas (mPFC, NAc, Amygdala, LS) is interesting. Are there any references that previous demonstrated this multi-target projection?

Yes, this subpopulation targeting 4 areas, together with other subpopulations, were reported in a very recent study (Gergues et al., 2020).

The vCA1 neurons connect with a wide range of brain areas including mPFC, NAc, amygdala, LS, hypothalamus, thalamus and entorhinal cortex etc (Cenquizca and Swanson, 2007; Pitkanen et al., 2000). There have been studies showing different subpopulations of hippocampal cells project to distinct downstream targets and exert different functions: BA-projecting neurons for contextual fear, hypothalamus-projecting neurons for anxiety and NAc-projecting neurons for cocaine conditioned place preference (Jimenez et al., 2018; Xu et al., 2016; Zhou et al., 2019). Interestingly, hippocampal neurons with distinct projection patterns appear to be mediating separable physiological functions (Ciocchi et al., 2015). For example, mPFC-projecting neurons are selectively activated for anxiety, NAc-projecting neurons are selectively activated for goal-directed behavior and triple-projecting neurons, targeting the mPFC, amygdala, and NAc, are selectively activated for sharp wave/ripples. Recent studies demonstrated that there are more complicated projection patterns for multiple brain areas, including the subpopulation targeting 4 brain areas of mPFC, NAc, Amygdala and LS (Gergues et al., 2020). We chose this subpopulation as a starting point to look into the response of vCA1 neurons to emotional stimuli. We hope to extend the application of the IBIST system to neurons with other projection patterns and dissect the circuit mechanism for such distinct emotional processing in the vCA1. The related discussion was updated accordingly (**page 16, Line 388-400**).

7. It would be informative for readers to have a discussion about the differences between the INTERSECT method from Deisseroth lab (Fenno et al 2020) and IBIST, perhaps in terms

of efficiency and expression levels, ease of vector construction, also the leakiness of TRE vectors.

Thanks for the suggestion. In response to this comment, please see our discussion about the differences between the two systems at our reply to point 5# from reviewer 2#. In brief, the commonality of effector vectors in the IBIST system is higher than that in the INTRSECT system. The vector construction in the IBIST system is easier because the DNA lengths of tTAN and tTAC are shorter in the IBIST system and they are separated in two constructs. The IBIST system is able to intersect more features than the INTRSECT system. We are cautious about the potential leakiness of TRE effectors, but we have verified some good TRE effectors and we expect to see a list of qualified TRE vectors in the research community shared with each other. We have also updated relevant discussion in the text (page 15, line 347-353, 362-364).

Reference:

- Bedbrook, C.N., Yang, K.K., Robinson, J.E., Mackey, E.D., Gradinaru, V., and Arnold, F.H. (2019). Machine learning-guided channelrhodopsin engineering enables minimally invasive optogenetics. *Nat Methods* *16*, 1176-1184.
- Cassel, J.C., Pereira de Vasconcelos, A., Loureiro, M., Cholvin, T., Dalrymple-Alford, J.C., and Vertes, R.P. (2013). The reuniens and rhomboid nuclei: neuroanatomy, electrophysiological characteristics and behavioral implications. *Prog Neurobiol* *111*, 34-52.
- Cenquizca, L.A., and Swanson, L.W. (2007). Spatial organization of direct hippocampal field CA1 axonal projections to the rest of the cerebral cortex. *Brain research reviews* *56*, 1-26.
- Ciocchi, S., Passecker, J., Malagon-Vina, H., Mikus, N., and Klausberger, T. (2015). Brain computation. Selective information routing by ventral hippocampal CA1 projection neurons. *Science* *348*, 560-563.
- Fenno, L.E., Ramakrishnan, C., Kim, Y.S., Evans, K.E., Lo, M., Vesuna, S., Inoue, M., Cheung, K.Y.M., Yuen, E., Pichamoorthy, N., *et al.* (2020). Comprehensive Dual- and Triple-Feature Intersectional Single-Vector Delivery of Diverse Functional Payloads to Cells of Behaving Mammals. *Neuron*.
- Gergues, M.M., Han, K.J., Choi, H.S., Brown, B., Clausing, K.J., Turner, V.S., Vainchtein, I.D., Molofsky, A.V., and Kheirbek, M.A. (2020). Circuit and molecular architecture of a ventral hippocampal network. *Nat Neurosci* *23*, 1444-1452.
- Jimenez, J.C., Su, K., Goldberg, A.R., Luna, V.M., Biane, J.S., Ordek, G., Zhou, P., Ong, S.K., Wright, M.A., Zweifel, L., *et al.* (2018). Anxiety Cells in a Hippocampal-Hypothalamic Circuit. *Neuron* *97*, 670-683 e676.

Miyamichi, K., Amat, F., Moussavi, F., Wang, C., Wickersham, I., Wall, N.R., Taniguchi, H., Tasic, B., Huang, Z.J., He, Z., *et al.* (2011). Cortical representations of olfactory input by trans-synaptic tracing. *Nature* 472, 191-196.

Olenych, S.G., Claxton, N.S., Ottenberg, G.K., and Davidson, M.W. (2007). The fluorescent protein color palette. *Curr Protoc Cell Biol Chapter 21*, Unit 21 25.

Patterson, G.H., Knobel, S.M., Sharif, W.D., Kain, S.R., and Piston, D.W. (1997). Use of the green fluorescent protein and its mutants in quantitative fluorescence microscopy. *Biophys J* 73, 2782-2790.

Pitkanen, A., Pikkarainen, M., Nurminen, N., and Ylinen, A. (2000). Reciprocal connections between the amygdala and the hippocampal formation, perirhinal cortex, and postrhinal cortex in rat. A review. *Annals of the New York Academy of Sciences* 911, 369-391.

Vila-Perello, M., and Muir, T.W. (2010). Biological applications of protein splicing. *Cell* 143, 191-200.

Xu, C., Krabbe, S., Grundemann, J., Botta, P., Fadok, J.P., Osakada, F., Saur, D., Grewe, B.F., Schnitzer, M.J., Callaway, E.M., *et al.* (2016). Distinct Hippocampal Pathways Mediate Dissociable Roles of Context in Memory Retrieval. *Cell* 167, 961-972 e916.

Zhou, Y., Zhu, H., Liu, Z., Chen, X., Su, X., Ma, C., Tian, Z., Huang, B., Yan, E., Liu, X., *et al.* (2019). A ventral CA1 to nucleus accumbens core engram circuit mediates conditioned place preference for cocaine. *Nat Neurosci* 22, 1986-1999.

REVIEWER COMMENTS

Reviewer #1 (Remarks to the Author):

In the revised manuscript titled, “An intein-split transactivator for intersectional neural imaging and optogenetic manipulation”, Chen and colleagues made extensive efforts to address most of the previously raised points. The revised manuscript is significantly improved, smoother to read, and more consistent overall. It is now able to highlight the validity as well as the applicability of the IBIST system. However, some remaining concerns, particularly with the way the analyses are conducted with the calcium imaging experiments and their interpretations prevent it from being seen as a finished product.

Major concerns:

1. Please provide low magnification images for the nucleus of reunien (RE) in figure S5k with nearby anatomical landmarks.

1a). The top leftmost (tTAN/mCherry) and rightmost (tTAN/tTAC/mCherry) panels seem to show different regions given the differences in the size and shape of the black spots. Please provide an explanation and/or better representative images.

1b). If the black spots are the third ventricle (which visually seems the case especially in the top rightmost panel), the dotted area is not the RE, which anatomically should be dorsal to the third ventricle rather than lateral. Please provide an explanation.

2. Throughout the calcium imaging experiments (figure 3-5), the authors made statements such as “significant increase in Ca²⁺ signals when animals were entering the center of the open field (line 258-259)”. These were made without the appropriate analyses. The authors always compared the magnitude of change in calcium activity before and after a cue (in this case, center entry) among different groups of animals. However, this type of analysis, on its own is not sufficient to make the abovementioned statement. To do so, one must compare the activity before the cue and after to examine whether a significant change in activity is observed. This is especially problematic in figure 4h. Comparing the magnitude of responses to emotionally relevant cues (water, sucrose, etc) among different CA1 populations with distinct projection profiles does not necessarily provide evidence that a certain of population is responsive to a certain cue, but only provide evidence of how different these responses are to each other. The comparisons are not sufficient to make the interpretations highlighted in lines 302-312.

2a). Secondary to this point, to make appropriate comparisons between activity before and after cue presentation, it is important to select baseline and cue windows with sufficient (for statistical comparison) and balanced number of timepoints. 0.5s baseline and 3s reward window for water/sucrose reward as well as 2s baseline and 10s response window for shock responses do not seem appropriate.

3. The authors made efforts to investigate the behavioral relevance of different CA1 populations with distinct projection and response profiles, particularly focusing on the mPFC/NAc and NAc/Amy dual projectors, which are selectively sensitive to footshocks only and to both footshocks and sucrose, respectively (figure 4). The authors then showed that the bidirectional manipulation of mPFC/NAc but not NAc/Amy dual projectors mediated real time place preference/avoidance (figure S8). Could the authors then further elaborate the function of NAc/Amy dual projectors? Do they behave as a salience detector?

Minor concerns:

1. In line 50-51, the authors stated the “requirement of high-level gene expression for optogenetic opsins and Ca²⁺ indicators”. Please provide some references to support the notion that high gene expression is necessary to express optogenetic opsins and Ca²⁺ indicators.
2. In line 192-194, the authors stated “monitoring specific population of neurons defined by intersectional conditions is extremely useful to dissect neuronal functions but also require high expression of Ca²⁺ indicators in vivo”. Please provide some references supporting the notion that high expression of Ca²⁺ indicators is necessary to study neuronal activity dynamics.
3. In line 216-218, the authors stated that “the hippocampal dorsal CA3 (dCA3) cells send collaterals to both pyramidal cells and interneurons in ventral CA1 (vCA1)”. Please provide some references for this anatomical connectivity.
4. Please provide justification in the text as to why the visual cortex is tested in primates rather than the hippocampus, where the IBIST tools are tested throughout in mice and across the vast majority of the manuscript.
5. Please provide justifications in the text why the PV- expressing CA1 cells with dCA3 inputs are tested for the expression of calcium indicators using the IBIST system (figure S5d-f) and SOM- expressing CA1 cells with dCA3 inputs are tested for NpHR expression (figure S6g-j), rather than just testing both conditions in the same cell type.
6. In the section (“IBIST-based Ca²⁺ imaging for cells with multiple projections) starting from line 280, the authors provided justifications for investigating the vCA1 neurons projecting to the mPFC and amygdala (line 285-286). Please also provide justification here why NAc and LS projectors are also investigated.

Reviewer #2 (Remarks to the Author):

The authors have addressed all my concerns satisfactorily, except the issue regarding cell viability and health. The authors counted the number of cells expressing the the pan-neuronal marker NeuN with and without expression of their system as a proxy for cell viability. This is not an adequate measure for cell viability, as only very large effects cell death would be found. Please measure standard apoptotic and necrotic markers with antibody stains (against caspase for example), or other applicable methods in cells with and without expression of the relevant viral vectors.

Reviewer #3 (Remarks to the Author):

The revised manuscript addressed my previous concerns. There are many "Figures for Reviewers" in the response letter that are not in the actual manuscript, which is frustrating when I read the revised manuscript first. I strongly recommend the authors putting "figures and tables for reviewers" into the manuscript (they can be supplemental figures panels and supplemental tables). Overall, I think this is a useful method, and with all the plasmids deposited in Addgene, people will be interested in using this tool kit.

Response to Reviewers

We thank the reviewers for their constructive comments and positive feedbacks. To address the reviewer comments, we have conducted additional experiments and analyses and have revised the manuscript as described below.

Reviewer comments are shown in **BLACK**, our responses are shown in **BLUE** and manuscript text changes are shown in **RED**.

Reviewer #1:

In the revised manuscript titled, “An intein-split transactivator for intersectional neural imaging and optogenetic manipulation”, Chen and colleagues made extensive efforts to address most of the previously raised points. The revised manuscript is significantly improved, smoother to read, and more consistent overall. It is now able to highlight the validity as well as the applicability of the IBIST system. However, some remaining concerns, particularly with the way the analyses are conducted with the calcium imaging experiments and their interpretations prevent it from being seen as a finished product.

We thank the reviewer for encouraging comments.

Major concerns:

1. Please provide low magnification images for the nucleus of reuniens (RE) in figure S5k with nearby anatomical landmarks.

1a). The top leftmost (tTAN/mCherry) and rightmost (tTAN/tTAC/mCherry) panels seem to show different regions given the differences in the size and shape of the black spots. Please provide an explanation and/or better representative images.

1b). If the black spots are the third ventricle (which visually seems the case especially in the top rightmost panel), the dotted area is not the RE, which anatomically should be dorsal to the third ventricle rather than lateral. Please provide an explanation.

We agree with the reviewer that prior images were not representative for RE. We have followed the suggestion by providing new images of RE of low magnification in **Figure S5k**. Thank you.

2. Throughout the calcium imaging experiments (figure 3-5), the authors made statements such as “significant increase in Ca²⁺ signals when animals were entering the center of the open field (line 258-259)”. These were made without the appropriate analyses. The authors always compared the magnitude of change in calcium activity before and after a cue (in this case, center entry) among different groups of animals. However, this type of analysis, on its own is not sufficient to make the abovementioned statement. To do so, one must compare the activity before the cue and after to examine whether a significant change in activity is observed. This is especially problematic in figure

4h. Comparing the magnitude of responses to emotionally relevant cues (water, sucrose, etc) among different CA1 populations with distinct projection profiles does not necessarily provide evidence that a certain of population is responsive to a certain cue, but only provide evidence of how different these responses are to each other. The comparisons are not sufficient to make the interpretations highlighted in lines 302-312.

We agree with the reviewer that more rigorous analysis should be used to determine whether the photometric signals were significantly different from the baseline. To follow the reviewer's suggestion, we have updated the analysis of Ca^{2+} signals in Fig 3 – 5. If the Ca^{2+} signals in the response window is not significantly different from the baseline (Mann Whitney U Test), the amplitude of Ca^{2+} response in this animal would be set as zero before averaging the results from a group of animals. We thank the reviewer for this suggestion and it indeed helped to clear the small but insignificant responses which might result from the fluctuation of Ca^{2+} signals.

We have updated all the analysis for Ca^{2+} signals in Fig 3 – 5 wherever applicable and the methods description (**line 1271 - 1273**). Importantly, these updated results still led to the same conclusions as before.

2a). Secondary to this point, to make appropriate comparisons between activity before and after cue presentation, it is important to select baseline and cue windows with sufficient (for statistical comparison) and balanced number of timepoints. 0.5s baseline and 3s reward window for water/sucrose reward as well as 2s baseline and 10s response window for shock responses do not seem appropriate.

We agree with the reviewer that it is important to have balanced number of timepoints for statistical comparison. We have used longer baseline wherever possible such that the baseline has the same time points as the response window. In some experiments, the inter-trial intervals for water/sucrose reward are very small because of the animal's high motivation. Therefore, it is not possible to define the baseline and response windows with the same time points. To address this concern, we divided the baseline and response periods into equal numbers of bins and averaged the Ca^{2+} signals in each bin (adapted from Fadok et al., 2017 Fig.3i and methods). We then determined whether the Ca^{2+} signals in the response window is significantly different from the baseline based on these balanced and binned data. The analyses are all updated in the manuscript and all support the same conclusions as before. Thanks to the reviewer's suggestion, we think that these updated analyses have made our conclusions more convincing (methods updated at **line 1271 – 1273**).

3. The authors made efforts to investigate the behavioral relevance of different CA1 populations with distinct projection and response profiles, particularly focusing on the mPFC/NAc and NAc/Amy dual projectors, which are selectively sensitive to footshocks only and to both footshocks and sucrose, respectively (figure 4). The authors then showed

that the bidirectional manipulation of mPFC/NAc but not NAc/Amy dual projectors mediated real time place preference/avoidance (figure S8). Could the authors then further elaborate the function of NAc/Amy dual projectors? Do they behave as a salience detector?

Thanks for this insightful thinking. Our Ca²⁺ recording showed that the NAc/Amy dual projectors responded to all rewarding and aversive stimuli in our experiments. Thus, we do think these projectors function as salience detectors for different kinds of emotional stimuli. This can be another unreported form of physiological functions of ventral hippocampal neurons.

We have added this discussion into the manuscript (**line 402 - 404**).

Minor concerns:

1. In line 50-51, the authors stated the “requirement of high-level gene expression for optogenetic opsins and Ca²⁺ indicators”. Please provide some references to support the notion that high gene expression is necessary to express optogenetic opsins and Ca²⁺ indicators.

The requirement of high-level gene expression for optogenetic opsins and Ca²⁺ indicators is well acknowledged in the research community (Chen et al., 2012; Dana et al., 2014; Deisseroth, 2015). As such, it is quite challenging to achieve high-level gene expression in transgenic mice. To date, the limited success is based on the use of the Thy1.2 promoter in randomly integrated transgenes (Arenkiel et al., 2007; Dana et al., 2014; Feng et al., 2000; Zhao et al., 2008). For intersectional control, there are only a handful of reported mouse lines using genomic locus TIGRE (Madisen et al., 2015). As a result, most of current approaches for opsins and indicators rely on viral methods to achieve high-level gene expression.

We thank the reviewer’s suggestion and have added these references (Deisseroth, 2015; Madisen et al., 2015) in the manuscript (**line 51**).

2. In line 192-194, the authors stated “monitoring specific population of neurons defined by intersectional conditions is extremely useful to dissect neuronal functions but also require high expression of Ca²⁺ indicators in vivo”. Please provide some references supporting the notion that high expression of Ca²⁺ indicators is necessary to study neuronal activity dynamics.

This is addressed in the previous answer. We have provided the references (Chen et al., 2012; Dana et al., 2014) in the manuscript as suggested (**line 195**).

3. In line 216-218, the authors stated that “the hippocampal dorsal CA3 (dCA3) cells send collaterals to both pyramidal cells and interneurons in ventral CA1 (vCA1)”. Please provide some references for this anatomical connectivity.

There are reports showing that the feed-forward inhibition onto CA1 pyramidal neurons after stimulating Schaffer collaterals originated from CA3 (Jang et al., 2015; Willadt et al., 2013; Zemankovics et al., 2013). Thus, the CA3 neurons target both pyramidal cells and interneurons in CA1.

We have added these references in the revised manuscript (**line 219**).

4. Please provide justification in the text as to why the visual cortex is tested in primates rather than the hippocampus, where the IBIST tools are tested throughout in mice and across the vast majority of the manuscript.

We chose the visual cortex in the primate for practical reasons. The visual cortex is in the surface of the brain and is easily accessible during the surgery. The hippocampus is deep in the primate brain, thus there are higher risks of off targeting for the AAV injection. As only one primate animal is usable in our study, we chose the safest way to test in the cortex.

5. Please provide justifications in the text why the PV- expressing CA1 cells with dCA3 inputs are tested for the expression of calcium indicators using the IBIST system (figure S5d-f) and SOM- expressing CA1 cells with dCA3 inputs are tested for NpHR expression (figure S6g-j), rather than just testing both conditions in the same cell type.

In the manuscript of our first submission, the Ca^{2+} indicator and NpHR were both tested in SOM+ cells. However, we could not adequately access the specificity and efficiency of the IBIST system in SOM+ cells because of the unsuccessful immunostaining of SOM. In the revised manuscript, we investigated the specificity and efficiency of the IBIST system in PV+ cells because we managed to get good immunostaining of PV. Therefore, this would help to address the reviewer comments and better access the specificity of the IBIST system.

6. In the section (“IBIST-based Ca^{2+} imaging for cells with multiple projections) starting from line 280, the authors provided justifications for investigating the vCA1 neurons projecting to the mPFC and amygdala (line 285-286). Please also provide justification here why NAc and LS projectors are also investigated.

There are ample evidence showing that vCA1 projectors have distinct physiological functions depending on their targets. For example, the NAc projector contributed to drug-induced place preference (Zhou et al., 2019) and social memory (Okuyama et al., 2016). The amygdala projector contributed to fear conditioning (Xu et al., 2016) and the LS projector contributed to feeding (Sweeney and Yang, 2015). Recent studies also showed that different subgroups of vCA1 neurons connected with distinct patterns of downstream areas (Gergues et al., 2020) and exhibited distinct physiological functions (Ciochi et al., 2015).

To follow the reviewer's suggestion, we modified the text and provided more specific information on the function of NAc and LS projectors (line 284 – 287).

Reviewer #2:

The authors have addressed all my concerns satisfactorily, except the issue regarding cell viability and health. The authors counted the number of cells expressing the the pan-neuronal marker NeuN with and without expression of their system as a proxy for cell viability. This is not an adequate measure for cell viability, as only very large effects cell death would be found. Please measure standard apoptotic and necrotic markers with antibody stains (against caspase for example), or other applicable methods in cells with and without expression of the relevant viral vectors.

We appreciate these comments. To address the reviewer's concern, we followed the suggestion and performed immunostaining of caspase to assess the cell viability (Fig. R1a and b). In the brain slices six weeks post IBIST infection, which is sufficiently long enough for most functional experiments, we found that nearly all IBIST-infected cells were free of Caspase signals (Fig. R1c). When comparing FOV fluorescent intensities in the brain slices (same acquisition settings), those for IBIST infected slices were very similar to those for negative control slices (saline injection), but were sharply different from those in positive control slices (Fig. R1d). These results demonstrated that IBIST-infected cells were viable and healthy enough for functional study.

Figure Rebuttal 1. Caspase immunostaining of brain slices with IBIST expression. (a) Scheme depicting AAV injections into hippocampus for IBIST expression, positive control (AAV-Caspase3) and negative control

(Saline). **(b)** Examples showing Caspase3 signals and mCherry signals in the hippocampal slices from saline (negative control), Caspase3 (positive control) and IBIST groups, respectively. For the IBIST slice, right pictures are zoom in of square areas in left ones. Arrows indicates detectable but weak Caspase signals. Scale bars: 100 μ m. **(c)** Percentage summary of mCherry cells with detectable Caspase3 ($0.44 \pm 0.23\%$, $n = 24$ FOV, $N = 6$ animals). **(d)** Summary of FOV fluorescent intensities in brain slices from animals injected with saline (negative control), IBIST vectors and Caspase3 vectors (positive control). One-way ANOVA revealed significant fluorescence differences between groups ($F_{(2, 39)} = 60.2$, $P = 1.2 \times 10^{-12}$) and Turkey's multiple comparisons revealed that the fluorescence is significantly higher in AAV-Caspase3 group than in others (Saline vs. AAV-Caspase3, $P = 1.97 \times 10^{-12}$; IBIST vs. AAV-Caspase3, $P = 1.28 \times 10^{-11}$). Saline, $n = 12$ FOVs, $N = 5$ animals; IBIST, $n = 24$ FOVs, $N = 6$ animals; AAV-Caspase3, $n = 6$ FOVs, $N = 2$ animals.

Reviewer #3:

The revised manuscript addressed my previous concerns. There are many "Figures for Reviewers" in the response letter that are not in the actual manuscript, which is frustrating when I read the revised manuscript first. I strongly recommend the authors putting "figures and tables for reviewers" into the manuscript (they can be supplemental figures panels and supplemental tables). Overall, I think this is a useful method, and with all the plasmids deposited in Addgene, people will be interested in using this tool kit.

We appreciate these encouraging comments. Thank you.

To follow the reviewer's suggestion, we have moved the following "figures and tables for reviewers" into the revised manuscript.

Prior rebuttal Figure 4 was inserted in the Supplementary Figure 2d – f.

Prior rebuttal Figure 7 was inserted in the Supplementary Figure 7.

Prior rebuttal Table 1 was inserted in the Supplementary Table 1.

The text is also updated accordingly (**line 168 – 170**, **line 289 – 290**).

Reference:

Arenkiel, B.R., Peca, J., Davison, I.G., Feliciano, C., Deisseroth, K., Augustine, G.J., Ehlers, M.D., and Feng, G. (2007). In vivo light-induced activation of neural circuitry in transgenic mice expressing channelrhodopsin-2. *Neuron* 54, 205-218.

Chen, Q., Cichon, J., Wang, W., Qiu, L., Lee, S.J., Campbell, N.R., Destefino, N., Goard, M.J., Fu, Z., Yasuda, R., *et al.* (2012). Imaging neural activity using Thy1-GCaMP transgenic mice. *Neuron* 76, 297-308.

Ciocchi, S., Passecker, J., Malagon-Vina, H., Mikus, N., and Klausberger, T. (2015). Brain computation. Selective information routing by ventral hippocampal CA1 projection neurons. *Science* 348, 560-563.

Dana, H., Chen, T.W., Hu, A., Shields, B.C., Guo, C., Looger, L.L., Kim, D.S., and Svoboda, K. (2014). Thy1-GCaMP6 transgenic mice for neuronal population imaging in vivo. *PLoS One* 9, e108697.

Deisseroth, K. (2015). Optogenetics: 10 years of microbial opsins in neuroscience. *Nat Neurosci* 18, 1213-1225.

Fadok, J.P., Krabbe, S., Markovic, M., Courtin, J., Xu, C., Massi, L., Botta, P., Bylund, K., Muller, C., Kovacevic, A., *et al.* (2017). A competitive inhibitory circuit for selection of active and passive fear responses. *Nature* 542, 96-100.

Feng, G., Mellor, R.H., Bernstein, M., Keller-Peck, C., Nguyen, Q.T., Wallace, M., Nerbonne, J.M., Lichtman, J.W., and Sanes, J.R. (2000). Imaging neuronal subsets in transgenic mice expressing multiple spectral variants of GFP. *Neuron* 28, 41-51.

Gergues, M.M., Han, K.J., Choi, H.S., Brown, B., Clausing, K.J., Turner, V.S., Vainchtein, I.D., Molofsky, A.V., and Kheirbek, M.A. (2020). Circuit and molecular architecture of a ventral hippocampal network. *Nat Neurosci* 23, 1444-1452.

Jang, H.J., Park, K., Lee, J., Kim, H., Han, K.H., and Kwag, J. (2015). GABA_A receptor-mediated feedforward and feedback inhibition differentially modulate the gain and the neural code transformation in hippocampal CA1 pyramidal cells. *Neuropharmacology* 99, 177-186.

Madisen, L., Garner, A.R., Shimaoka, D., Chuong, A.S., Klapoetke, N.C., Li, L., van der Bourg, A., Niino, Y., Egolf, L., Monetti, C., *et al.* (2015). Transgenic mice for intersectional targeting of neural sensors and effectors with high specificity and performance. *Neuron* 85, 942-958.

Okuyama, T., Kitamura, T., Roy, D.S., Itohara, S., and Tonegawa, S. (2016). Ventral CA1 neurons store social memory. *Science* 353, 1536-1541.

Sweeney, P., and Yang, Y. (2015). An excitatory ventral hippocampus to lateral septum circuit that suppresses feeding. *Nat Commun* 6, 10188.

Willadt, S., Nenniger, M., and Vogt, K.E. (2013). Hippocampal feedforward inhibition focuses excitatory synaptic signals into distinct dendritic compartments. *PLoS One* 8, e80984.

Xu, C., Krabbe, S., Grundemann, J., Botta, P., Fadok, J.P., Osakada, F., Saur, D., Grewe, B.F., Schnitzer, M.J., Callaway, E.M., *et al.* (2016). Distinct Hippocampal Pathways Mediate Dissociable Roles of Context in Memory Retrieval. *Cell* 167, 961-972 e916.

Zemankovics, R., Veres, J.M., Oren, I., and Hajos, N. (2013). Feedforward inhibition underlies the propagation of cholinergically induced gamma oscillations from hippocampal CA3 to CA1. *J Neurosci* 33, 12337-12351.

Zhao, S., Cunha, C., Zhang, F., Liu, Q., Gloss, B., Deisseroth, K., Augustine, G.J., and Feng, G. (2008). Improved expression of halorhodopsin for light-induced silencing of neuronal activity. *Brain Cell Biol* 36, 141-154.

Zhou, Y., Zhu, H., Liu, Z., Chen, X., Su, X., Ma, C., Tian, Z., Huang, B., Yan, E., Liu, X., *et al.* (2019). A ventral CA1 to nucleus accumbens core engram circuit mediates conditioned place preference for cocaine. *Nat Neurosci* 22, 1986-1999.

REVIEWERS' COMMENTS

Reviewer #1 (Remarks to the Author):

After a second round of revision, Chen and colleagues provided a much improved manuscript titled “An intein-split transactivator for intersectional neural imaging and optogenetic manipulation” that addressed most of the previously raised concerns. While the authors should be applauded for their extensive efforts on updating the analyses regarding the calcium imaging experiments, balancing the number of timepoints by using different bin windows for the baseline and response windows, only superficially addresses the issue and have not dealt with the core. Having balanced number of timepoints and bin windows both are important for a fair statistical comparison between the baseline and response windows. The lack of gap periods between trials owing to an inadequate design is the reason why such compromises needed to be made for statistical comparisons afterwards, the author should carefully consider these points ahead of time in their next experimental design.

In supplementary figure 7c, the merged image does not match the images of the individual channels which it is supposedly composed of. For example, the red labeling (top row, supplementary figure 7c) inside the ROI is only visible in the merged image but not the image with the red channel on its own. Similar issues are observed in both the second and third rows of this figure.

Reviewer #2 (Remarks to the Author):

The authors have addressed my last concern and I support publication of this paper in its current form.

Response to Reviewers

We thank the reviewers for their positive feedbacks.

Reviewer comments are shown in **BLACK**, our responses are shown in **BLUE** and manuscript text changes are shown in **RED**.

Reviewer #1:

After a second round of revision, Chen and colleagues provided a much improved manuscript titled “An intein-split transactivator for intersectional neural imaging and optogenetic manipulation” that addressed most of the previously raised concerns. While the authors should be applauded for their extensive efforts on updating the analyses regarding the calcium imaging experiments, balancing the number of timepoints by using different bin windows for the baseline and response windows, only superficially addresses the issue and have not dealt with the core. Having balanced number of timepoints and bin windows both are important for a fair statistical comparison between the baseline and response windows. The lack of gap periods between trials owing to an inadequate design is the reason why such compromises needed to be made for statistical comparisons afterwards, the author should carefully consider these points ahead of time in their next experimental design.

We thank the reviewer for constructive comments. The point to consider fair statistical comparison in the experimental design is well taken. Thank you.

In supplementary figure 7c, the merged image does not match the images of the individual channels which it is supposedly composed of. For example, the red labeling (top row, supplementary figure 7c) inside the ROI is only visible in the merged image but not the image with the red channel on its own. Similar issues are observed in both the second and third rows of this figure.

Thank you for pointing this out. We had a careful examination of individual channels of the merged images. The red labeling observed by the referee does exist in the red channel but is not that obvious without the overlay of the DIC images. Consistently, we did not see any unmatched red labeling when only merging green and red channels. In the updated Supplementary figure 7, we have provided the DIC image separately and showed the merged pictures with green and red channels.

We appreciate the careful examination by the referee, which greatly helped us to improve the clarity and readability of the manuscript.

Reviewer #2:

The authors have addressed my last concern and I support publication of this paper in its current form.

Thank you very much for your support.